# Shore crabs reveal novel evolutionary attributes of the mushroom body

**Nicholas Strausfeld[1]\*, Marcel E Sayre[2,3]**

[1]Department of Neuroscience, University of Arizona, Tucson, United States; [2]Lund Vision Group, Department of Biology, Lund University, Lund, Sweden; [3]Department of Biological Sciences, Macquarie University, Sydney, New South Wales, Australia

**Abstract** Neural organization of mushroom bodies is largely consistent across insects, whereas the ancestral ground pattern diverges broadly across crustacean lineages resulting in successive loss of columns and the acquisition of domed centers retaining ancestral Hebbian-like networks and aminergic connections. We demonstrate here a major departure from this evolutionary trend in Brachyura, the most recent malacostracan lineage. In the shore crab *Hemigrapsus nudus*, instead of occupying the rostral surface of the lateral protocerebrum, mushroom body calyces are buried deep within it with their columns extending outwards to an expansive system of gyri on the brain's surface. The organization amongst mushroom body neurons reaches extreme elaboration throughout its constituent neuropils. The calyces, columns, and especially the gyri show DC0 immunoreactivity, an indicator of extensive circuits involved in learning and memory.

## Introduction

Insect mushroom bodies, particularly those of *Drosophila*, are the most accessible models for elucidating molecular and computational algorithms underlying learning and memory within genetically and connectomically defined circuits (e.g. *Aso et al., 2014a*; *Senapati et al., 2019*; *Jacob and Waddell, 2020*; *Modi et al., 2020*). An advantage of *Drosophila* is that it exemplifies a ground pattern organization that is largely consistent across hexapod lineages (*Li and Strausfeld, 1997*; *Ito et al., 1998*; *Ito et al., 2014*; *Sinakevitch et al., 2001*; *Groh and Rössler, 2011*; *Montgomery and Ott, 2015*).

Because of the importance of mushroom bodies in understanding the relevance of synaptic organization in sentience and cognition, recognizing evolutionary divergence of these centers would be expected to yield experimentally testable predictions about evolved modifications of circuitry in relation to ecological demands imposed on the species. However, insect mushroom bodies show a remarkably conserved organization, which makes them relatively unsuited for neuroevolutionary studies. There are some minor exceptions, to be sure: in honey bees and ants, calycal domains receive modality-specific afferents; in beetles, dietary generalists have more elaborate calyces than specialists; in aquatic beetles, a modality switch has replaced olfactory input with a visual input into the calyces (*Gronenberg, 2001*; *Farris and Schulmeister, 2011*; *Lin and Strausfeld, 2012*). Less attention has been given to neural arrangements comprising the mushroom body lobes (columns), although some studies have addressed distinctions in basal groups such as silverfish (Zygentoma), dragonflies and mayflies (*Farris, 2005*; *Strausfeld et al., 2009*).

The evolutionary stability of the insect mushroom body contrasts with the recent demonstration that mushroom bodies of malacostracan crustaceans have undergone substantial and often dramatic modification of the mandibulate (ancestral) mushroom body ground pattern (*Stegner and Richter, 2011*; *Wolff and Strausfeld, 2015*; *Wolff et al., 2017*; *Strausfeld et al., 2020*). Whereas Stomatopoda (mantis shrimps), the sister group of Eumalacostraca, possesses mushroom bodies that correspond to those of insects, equipped with calyces from which arise prominent columns (*Wolff et al.,*

**\*For correspondence:**
flybrain@neurobio.arizona.edu

**Competing interests:** The authors declare that no competing interests exist.

*2017*), the trend over geological time has been a reduction of those columnar components such that the ancestral ground pattern organization of parallel fibers and orthogonal Hebbian networks has morphed to provide planar arrangements within domed centers lacking columns (*Wolff and Strausfeld, 2015*; *Sayre and Strausfeld, 2019*; *Strausfeld et al., 2020*). These neuronal adaptations, nevertheless, share the property with insect mushroom bodies of being immunopositive to an antibody raised against the catalytic subunit of protein kinase A, encoded by the *Drosophila* gene DC0, that is required for effective learning and memory (*Kalderon and Rubin, 1988*; *Skoulakis et al., 1993*). Antibodies raised against DC0 are reliable identifiers of neuropils mediating learning and memory in arthropods and other phyla (*Strausfeld et al., 2009*; *Wolff et al., 2012*; *Wolff and Strausfeld, 2015*).

Whereas the evolutionary shift from columnar mushroom bodies to noncolumnar homologues is demonstrated across most decapod crustaceans, one lineage has until now defied unambiguous identification of a columnar or even a noncolumnar center. This is Brachyura, known in the vernacular as true crabs, here represented by *Hemigrapsus nudus*. Brachyura is a comparatively young lineage, recognized from fossils dating from the mid-Jurassic (*Schweitzer and Feldmann, 2010*; *Guinot, 2019*). Phylogenomics places the origin of Brachyura also as mid-to-late Jurassic (*Wolfe et al., 2019*). Brachyura is the most species-rich decapod clade, comprising 6793 currently known species (*Ng et al., 2008*). It is also a hugely successful lineage; extant species occupy benthic, littoral, estuarine, brackish, fresh water, terrestrial and even arboreal habitats. Many of these ecologies are defined by complex topographies (*Hartnell, 1988*; *Lee, 2015*).

Historically, identifying a mushroom body homologue in the crab's brain has been problematic. Claims for homologous centers range from paired neuropils in the brain's second segment, the deutocerebrum, later attributed to the olfactory system (*Bethe, 1897*), to an insistence that the crab's reniform body is a mushroom body (*Maza et al., 2016*; *Maza et al., 2021*). Demonstrated 138 years ago in stomatopod crustaceans, the reniform body is a morphologically distinct center that coexists in the brain's lateral protocerebrum adjacent to its columnar mushroom bodies (*Bellonci, 1882*; *Thoen et al., 2020*).

Until the present, observations of the varunid shore crab *Hemigrapsus nudus* have identified large anti-DC0-reactive domains occupying almost the entire rostral volume of its lateral protocerebrum but no evidence for mushroom bodies (*Thoen et al., 2020*). The affinity of these domains to anti-DC0 suggested, however, a cognitive center far more expansive than found in any other arthropod of comparable size, with the possible exception of the domed mushroom body of the land hermit crab *Coenobita clypeatus* (*Wolff et al., 2012*). Here we demonstrate that in the shore crab there are indeed paired mushroom bodies. But these have been 'hiding in plain sight,' having undergone an entirely unexpected neurological transformation that is opposite to the evolutionary trend toward a domed noncolumnar morphology shown in other lineages.

Here, we provide evidence at the level of neuronal arrangements showing that in the shore crab paired mushroom bodies have undergone an evolved transformation that is possibly unique to Arthropoda. They are inverted: their calyces reside deep within the lateral protocerebrum, a location allowing outward expansion of the mushroom body columns, which reach expanded cortex-like folds beneath the brain's surface. The entire disposition of the shore crab's mushroom bodies is opposite to that of any other crustacean, or insect, in which the calyces are situated under the rostral surface of the lateral protocerebrum with their columns extending downwards into deeper neuropil (*Strausfeld et al., 2009*; *Wolff et al., 2017*; *Sayre and Strausfeld, 2019*).

Traits defining neuronal organization demonstrate that the mushroom bodies of Stomatopoda phenotypically correspond to those defining the mushroom bodies of *Drosophila* (*Wolff et al., 2017*). The following description uses complementary methods that resolve those traits in *Hemigrapsus nudus*. Reduced silver staining demonstrates spatial arrangements of calycal and columnar neuropils and their detailed neuroarchitecture. Osmium-ethyl gallate treatment of intact brains resolves neuronal densities; and serial sections of these preparations provide the three-dimensional reconstructions shown throughout this account (see Materials and methods). Golgi mass-impregnations enable crucial insights into the exceptionally elaborate organization of mushroom body intrinsic neurons, identified as the phenotypic homologues of insect Kenyon cells. Immunohistology has been used to resolve DC0-immunopositive components of the mushroom body calyces and columns, and antibodies raised against GAD (glutamic acid decarboxylase) reveal putative levels of local inhibition. Antibodies raised against tyrosine hydroxylase (TH) and 5-hydroxytryptophan (5HT) demonstrate

arborizations consistent with those of output and input neurons intersecting different levels of the *Drosophila* and stomatopod mushroom body columns.

We show that despite the varunid mushroom body's unparalleled intricacy and its unique transformation of the ancestral ground pattern, its defining traits nevertheless demonstrate phenotypic homology with the mushroom bodies of insects and those of other crustaceans. We close with a Discussion proposing that the evolution of such intricacy, and the expansion of the lateral protocerebrum to accommodate mushroom body enlargement, is likely to have contributed novel circuits and, ultimately, to cognitive flexibility permitting varunid Brachyura to exploit complex ecological topographies at the interface of marine and terrestrial biotopes.

## Results

The Results are organized into eight sections, the first describing the overall location and structure of the varunid mushroom body. This is followed by the identification of its neural components and their arrangements as elaborate networks comprising the paired calyces. Descriptions of the internal organization of the calyces' rostral extensions, and their two prominent columns, culminates with an explanation of the cortex-like organization of overlying gyri to which the columns project. The final observations address the relationship of the mushroom body to the reniform body, a center common to malacostracan crustaceans. To provide a three-dimensional understanding of this highly intricate system, descriptions refer to locations and arrangements within a reconstructed lateral protocerebrum and its major components (see Materials and methods).

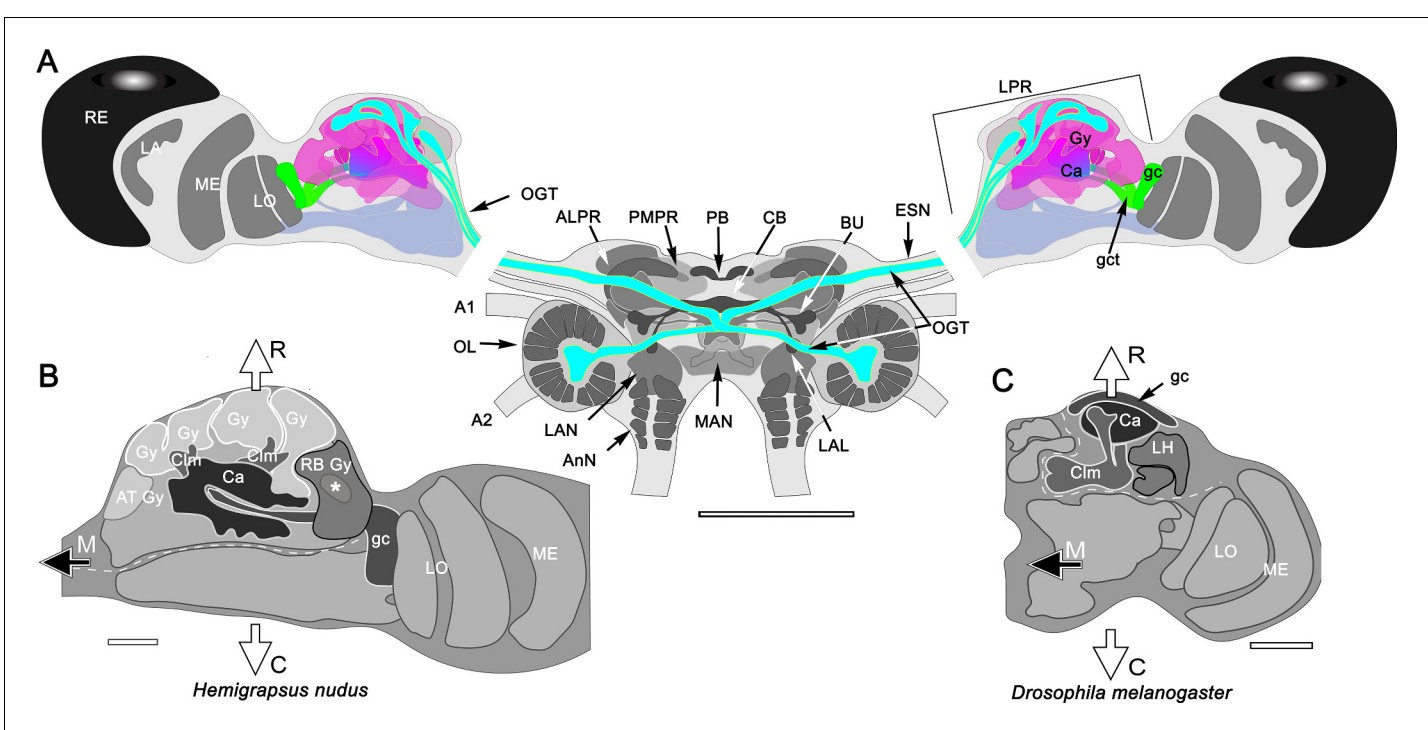

**Figure 1.** The brain of *Hemigrapsus nudus* and a comparison of its lateral protocerebrum with that of the fly *Drosophila melanogaster*. (**A**) Schematic of the *Hemigrapsus* brain (frontal view) and major neuropils. Abbreviations: ALPR, anterior lateral protocerebrum; A1, antennular nerve; A2, antennal nerve; AnN, antenna 2 neuropil; BU, lateral bulb of the central complex; CB, central body; ESN, eyestalk nerve; LAL, lateral accessory lobe; LAN, lateral antennular lobe neuropil; LA, lamina; LO, lobula; LPR, lateral protocerebrum; MAN, median antenna 1 neuropil; MB, mushroom body; ME, medulla; OL, olfactory lobe; OGT, olfactory globular tract; PMPR, posterior medial protocerebral neuropils; PB, protocerebral bridge; RE, compound retina. The eyestalk nerve (ESN) carries *all* axons connecting the brain with the lateral protocerebrum. (**B, C**) The crab lateral protocerebrum aligned with that of *Drosophila* (in both, the optic lobe's lamina is omitted) The white dashed line in *H. nudus* divides its rostral lateral protocerebral volume from its caudal volume; in *Drosophila* the dashed line divides the rostral volume of the protocerebrum from the rest of the hemi-brain. In both taxa, the filled arrow M indicates medial, and the open arrows R, C indicate, respectively, rostral and caudal. Both schematics depict a view from the ventral side of the lateral protocerebrum/ hemi-brain. Abbreviations: Ca, calyx; Clm, column; gc, globuli cells; gct, globuli cell tract; Gy, gyri; LH, lateral horn; LO, lobula complex; ME, medulla; RB, reniform body. Scale bars: **A**, 500 µm; **B**, 100 µm; **C**, 50 µm.

## The varunid lateral protocerebrum (*Figures 1–3*)

In crabs, as is typical of stomatopods and most decapod crustaceans, the most anterior part of its brain, the protocerebrum, is flanked each side by a lateral outgrowth that expands into an enlargement of the eyestalk immediately proximal to the retina (*Figure 1A*). This volume of brain comprises the lateral protocerebrum (*Figures 1A* and *2A*). The lateral protocerebrum is connected to the midbrain proper by the eyestalk nerve composed of axons relaying information to and from the midbrain.

Brains treated with osmium-ethyl gallate or with reduced silver (*Figures 2A* and *3*) resolve the varunid lateral protocerebrum as a hump-shaped volume divided into a rostral and caudal part. The rostral lateral protocerebrum (RLPR), once referred to as the terminal medulla, a now defunct term (*Strausfeld, 2020*), is dominated by two centers: the mushroom bodies and the smaller reniform body. There are a small number of other satellite neuropils near the origin of the eyestalk nerve. The mushroom body is immediately recognizable by its intensely osmiophilic calyces from which arise columns that extend outwards, branching into a cortex-like neuropil composed of gyri (*Figure 2A*). These occupy the outermost volume of the RLPR. The reniform body comprises an independent system of interconnected neuropils situated laterally constrained to a volume of the RLPR that extends from its dorsal to its ventral surface (*Thoen et al., 2020*). In longitudinal sections of the RLPR parallel to its rostro-caudal plane, the reniform body's prominent axon bundle, the pedestal, is seen in cross-section (RB in *Figures 2* and *3*).

The crab's RLPR corresponds to that volume of the insect protocerebrum containing the mushroom body and its immediately associated neuropils, such as the lateral horn (*Figure 1B,C*). As in stomatopods and decapods, the crab's RLPR receives terminals of olfactory projection neurons originating from both the ipsi- and contralateral olfactory (antennular) lobes located in the brain's second segment, the deutocerebrum (*Figure 1*). The axons of more than a thousand projection neurons are organized into several discrete fascicles bundled together to provide the olfactory globular tract (OGT). The left and right OGTs extend superficially across the midbrain to meet just dorsal to the central body. There, each tract bifurcates to provide inputs to each lateral protocerebrum from both the ipsi- and contralateral olfactory lobes (*Figure 1*). The eyestalk nerve also carries other efferent axons to the lateral protocerebrum: these include relay neurons from other sensory neuropils, relays from the opposite lateral protocerebrum and optic lobes, and interneurons relaying from various midbrain neuropils.

The caudal volume of the lateral protocerebrum (CLPR) is mainly associated with outputs from the optic lobe neuropils (*Figure 2A*). Most of these originate from the lobula, each bundle of outgoing axons representing a population of one of the numerous morphological types of columnar neurons. Bundles segregate to a system of interconnected neuropils called optic glomeruli, as occurs in insects (*Okamura and Strausfeld, 2007*). Further relays from the glomeruli send information centrally to the midbrain via the eyestalk nerve. The nerve also carries to the midbrain the axons of efferent neurons from the lateral protocerebrum's rostral neuropil, from the mushroom bodies and their associated gyri and from CLPR neuropils associated with the optic glomeruli.

## Organization of DC0-positive volumes (*Figure 2*)

Densely osmiophilic calyces and their extended columns (*Figure 2A*) correspond to discrete neuropils showing high levels of immunoreactivity to antibodies raised against DC0 (*Figure 2B*). The fibroarchitecture of these DC0-positive volumes is revealed by labeling with antibodies against α-tubulin or by reduced silver (*Figure 2E*; *Figure 3A-C*), which provides correlative data for resolving and interpreting the arrangements of the mushroom body volumes in 3D reconstructions (*Figure 3—figure supplement 1*: see Materials and methods). Salient both in osmium-ethyl gallate (*Figures 2A* and *3D,E*) and reduced silver-stained brains (*Figure 3A,C*) is the layer of gyri that defines the outer level of the rostral lateral protocerebrum. Gyri are DC0-immunoreactive and are contiguous with each of the DC0-immunoreactive columns that extend outwards from the calyces. Gyri are resolved as pillowed folds and indentations (sulci: *Figure 2D,F*) comparable to the folded architecture of a mammalian cerebral cortex. This feature appears to be unique to Brachyura (see Discussion).

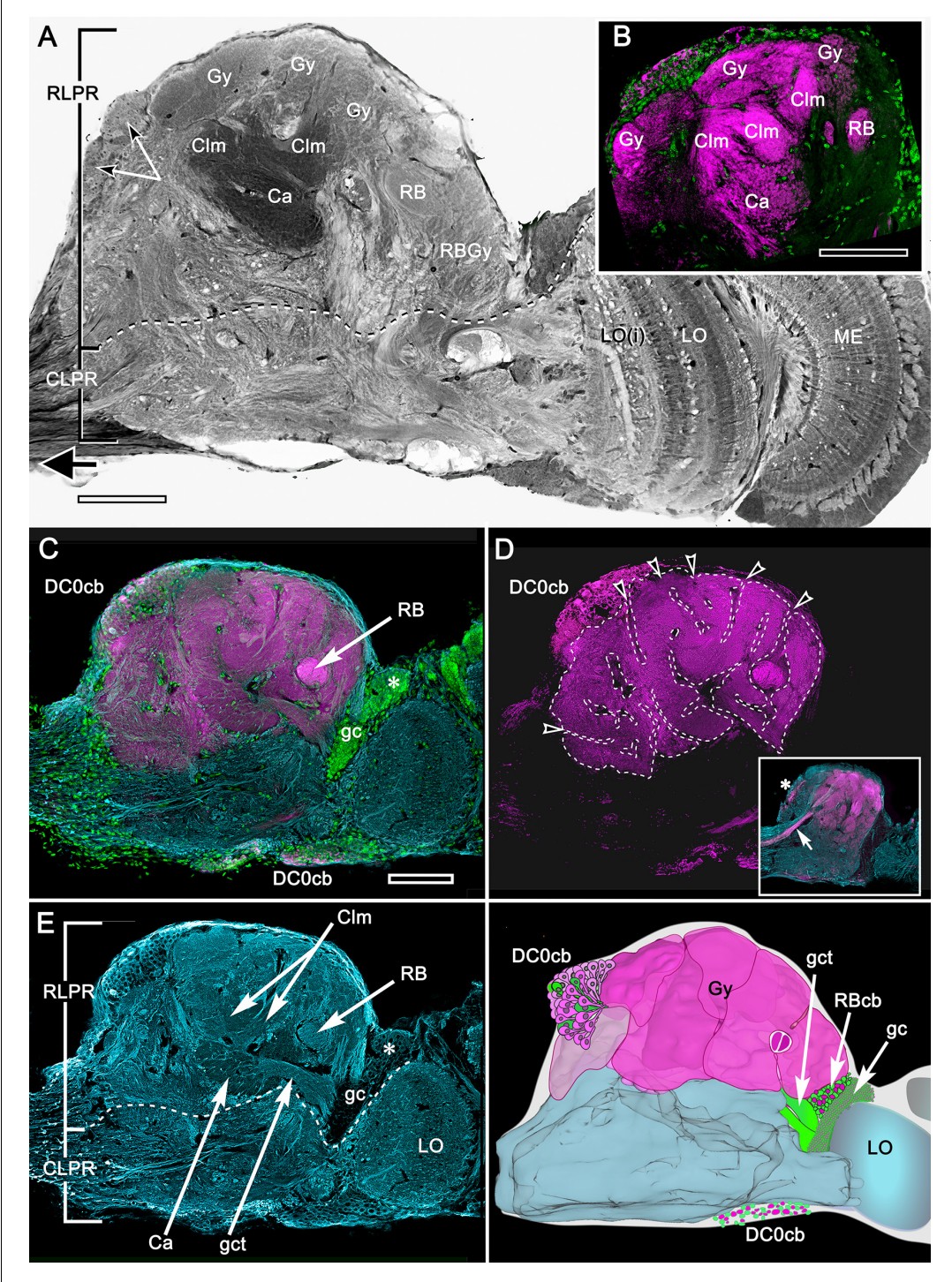

**Figure 2.** Organization of the varunid crab mushroom body and its associated gyri. (A) Brains treated with osmium-ethyl gallate demonstrate the overall disposition of neuropils, exemplified by this section of the lateral protocerebrum from its medial border to the optic lobe medulla. Its rostral domain (RLPR) includes the mushroom body calyces (Ca), columns (Clm) and their associated gyriform neuropils (Gy). Clusters of DC0-positive neuronal cell bodies (here DC0cb) are distributed over the neuropil. Distal volumes of the RLPR are associated with the reniform body (RB; here its pedestal and a gyrus RBGy), which is situated between the mushroom body and the optic lobe, here represented by its medulla (ME) and bilayered lobula (LO, LOi). The optic lobe provides outputs to discrete neuropils comprising the caudal lateral protocerebrum (CLPR). (B–E) Anti-DC0-labeled (magenta) volumes in panel B match the corresponding membrane-dense calyces and columns and their rostral gyri in panel A. Panel C demonstrates that DC0-immunostained gyri occupy almost the entire rostral level of the RLPR, with the exception of a medial gyrus (see *Figure 11*). Two small cell body

*Figure 2 continued on next page*

*Figure 2 continued*

clusters caudally (DC0cb) lining the CLPR, near the optic lobes, include DC0-immunoreactive perikarya, as does a prominent proximal cluster of large cell bodies (DC0cb; also indicated by arrows in panel **A**) immediately above the entry of the eyestalk nerve. The latter cell bodies belong to neurons that extend to gyri. DC0-immunoreactive axons extending from gyri into the eye stalk nerve (inset panel **D**, axons indicated by arrow, cell body cluster by asterisk) suggest that some of these DC0-immunoreactive neurons may constitute efferent channels to the midbrain. The nuclear stain Syto13 (green) demonstrates the location of mushroom body globuli cells (gc) lying immediately beneath a group of slightly larger perikarya (asterisk) that supply the reniform body. In panel **D**, removal of anti-α tubulin and Syto13 labeling from C resolves the extent of the gyri and their many sulcus-like indentations (open arrowheads). In panel **E**, anti-α-tubulin (cyan) shown here alone resolves the overall fibroarchitecture of the LPR to provide correlative data with those provided by reduced silver stains (*Figure 3*). (**F**) Total Amira-generated reconstruction of a serial-sectioned, osmium-ethyl gallate treated eyestalk demonstrates the huge area of the lateral protocerebrum occupied by gyri. Superimposed are the locations of small neurons supplying the reniform body (RBcb), mushroom body globuli cells (gc) and their tract of neurites leading to the calyx (gct), here obscured by the overlying gyri. Scale bars, **A**, **B**, 100 µm; **C-F**, 100 µm.

## The calyces and their origin from globuli cells (*Figures 2–4*)

If centers are claimed as phenotypic homologues of insect mushroom bodies, then those centers are expected to have the following categories of neurons irrespective of their divergence from the ancestral ground pattern as represented in the allotriocarid clade Cephalocarida (*Stegner and Richter, 2011*). These categories are: (1) intrinsic neurons, the most populous of which are Kenyon cells defining the mushroom body's volume, shape, and subdivisions; (2) extrinsic neurons providing inputs from sensory centers (*Kanzaki et al., 1989*; *Lin and Strausfeld, 2012*; *Li et al., 2020a*), as well as protocerebral regions encoding high-level multisensory information (*Li and Strausfeld, 1999*); and (3) output neurons that encode information computed by levels of the mushroom body lobes (*Li and Strausfeld, 1999*; *Turner et al., 2008*). The latter are referred to here as MBONs, adopting the term from studies on *Drosophila* (*Aso et al., 2014a*; *Aso et al., 2014b*).

Intrinsic neurons supplying the *Hemigrapsus nudus* calyces correspond to insect Kenyon cells. As do insect Kenyon cells, these originate from the brain's smallest and most crowded population of neuronal cell bodies, the globuli cells (*Figures 2C*, *3A,D* and *4*). In other pancrustaceans, globuli cell perikarya have been shown as dense clusters situated at the rostral surface of the lateral protocerebrum, usually near its proximal margin or immediately above the mushroom body calyces, as in *Drosophila* or Stomatopoda, or, in malacostracans with eyestalks, close to the lateral protocerebrum's attachment to the eyestalk nerve. In *H. nudus*, however, globuli cell perikarya are found in a completely different and surprising locality: a densely populated 'hidden' volume tucked between the proximal margin of the lobula and the most distal border of the lateral protocerebrum (*Figures 2C*, *3A* and *4A–C*). As in other pancrustaceans, these perikarya are the smallest perikarya of the brain, in the varunid ranging from 5 to 7 µm in diameter (compared in *Figure 3A*). In crabs used for these studies (carapace widths of 4–6 cm), we estimate that this perikaryal cluster accounts for at least 22,000 mushroom body intrinsic neurons shared by the two calyces (see Materials and methods). Cytological distinctions amongst cells in the cluster, denoted by their appearance after osmium-ethyl gallate staining (*Wigglesworth, 1957*) or reduced silver (*Figure 4D–G*), suggest that new globuli cells are being continuously generated from ganglion mother cells situated at the cluster's periphery (*Figure 4D–G*). Continuous generation of interneurons supplying the antennular lobes, and in the lateral protocerebrum, has been previously reported for the shore crab *Carcinus maenas* (*Schmidt, 1997*).

Cell body fibers (neurites) extending from the globuli cell cluster are tightly packed into two prominent bundles, called globuli cell tracts, one projecting obliquely ventrally the other obliquely dorsally to supply each of the two adjacent calyces (*Figure 3—figure supplement 1*). Neurites comprising these tracts fan out to supply each calyx with its ensembles of mostly orthogonally arranged dendritic networks (*Figure 3B*; *Figure 5—figure supplement 1*). The networks, which are resolved by silver stains, comprise overlapping populations of intrinsic neuron processes extending amongst a mosaic of many thousands of discrete microglomeruli (*Figures 3B* and *5A*). These are assumed to be sites of synaptic convergence (*Figure 5—figure supplement 1*), corresponding to microglomerular organization in the calyces of insect, stomatopod and caridean mushroom bodies (*Yasuyama et al., 2002*; *Sjöholm et al., 2006*; *Groh and Rössler, 2011*; *Wolff et al., 2017*; *Sayre and Strausfeld, 2019*). As described in the following section, components of microglomeruli include the terminals of afferent relays originating from the olfactory lobes, terminals from neuropils

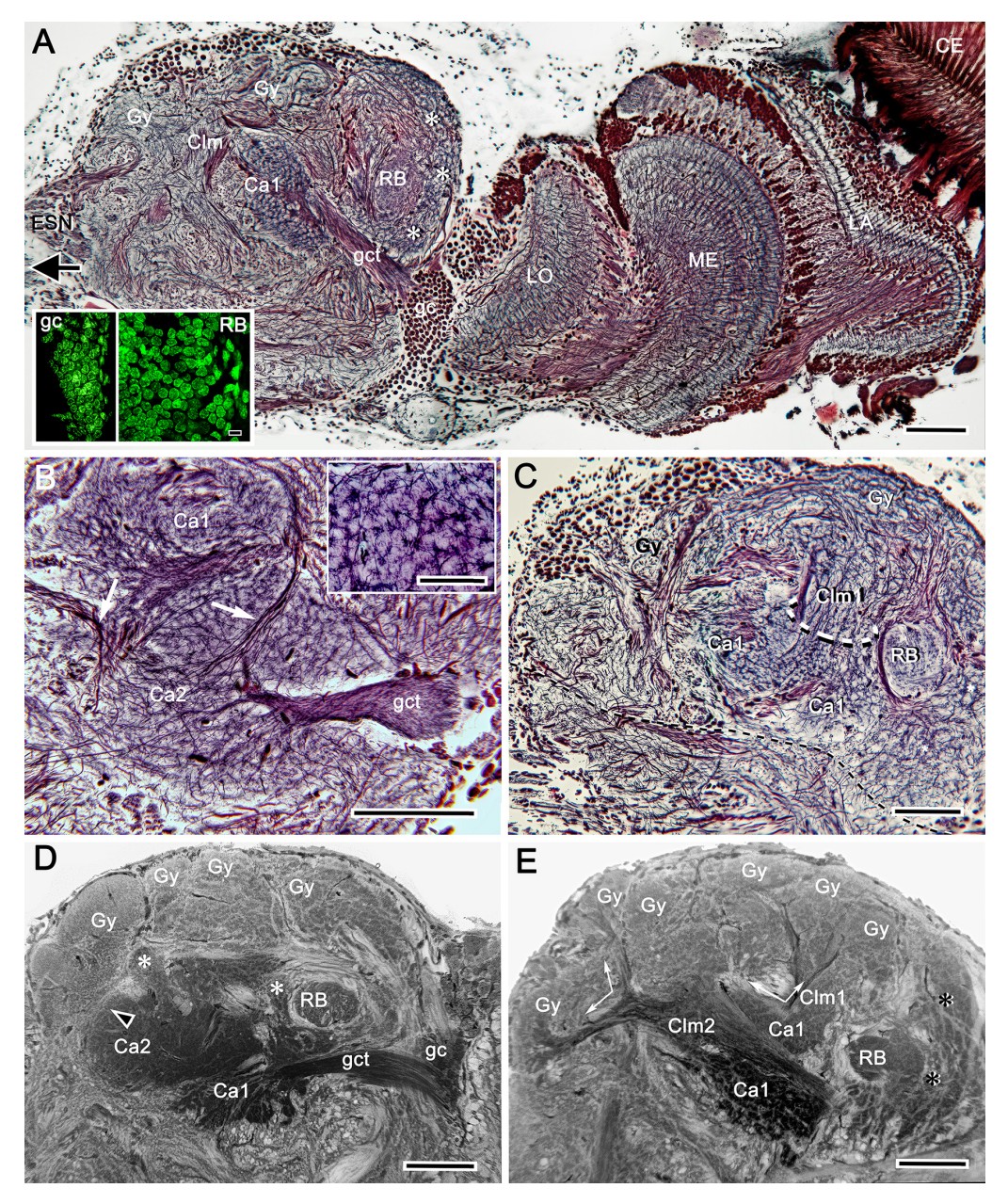

**Figure 3.** Uniquely identifiable neuropils of the mushroom bodies are resolved by reduced silver. (**A**) Longitudinal section parallel to the rostro-caudal plane of the lateral protocerebrum demonstrating the more dorsal of the two calyces (Ca1) and its supply by the globuli cell tract (gct) from the cluster of globuli cells (gc). Columnar extensions (Clm) from the calyces reach out to overlying gyri (Gy). The reniform body is shown with three of its gyri (asterisks) and its bundled axons (RB) situated between the mushroom body and optic lobes, insets lower left compare the smallest perikarya supplying the calyces (globuli cells, gc) with the next smallest perikarya belonging to neurons of the reniform body. Abbreviations: ESN, eyestalk nerve; CE, compound retina, LA, lamina, ME, medulla; LO, lobula. (**B**) Adjacent calyces (Ca1, Ca2) showing their diagnostic arrangement of microglomeruli (inset upper right; see also *Figure 5*). Arrows indicate afferent fibers into Ca2 from nonolfactory origins. (**C**) The calycal origin and rostral extension of the Ca1 column (Clm1, encircled). At this level, penetration by en passant axon bundles causes the calyx to appear fragmented. (**D, E**) Osmium ethyl gallate-treated sections at two depths. Panel **D** indicates the bulk and elaboration of calycal neuropils. At this level, calyx 2 appears as the larger. The arrowhead indicates the base of column 2. Calyx 1 is shown with its globuli cell tract entering a domain denoted by lateral extensions of calyx 1. In panel **E**, the base of column 1 (Clm1) is shown arising from Ca1, its curvature at this level splitting it into two parts. Column (Clm2) is shown extending two branches into the gyriform layer above. Clm1 extends outwards just proximal to the pedestal of the reniform body (RB). In addition to columns, calyces provide occasional narrower finger-like extensions extending towards the gyri (asterisks in D). Scale bars, 100 μm. Inset in **A**, 10 μm, inset in **B**, 50 μm.

*Figure 3 continued on next page*

*Figure 3 continued*

The online version of this article includes the following figure supplement(s) for figure 3:

**Figure supplement 1.** Globuli cells provide two tracts, one to each calyx as shown in this Amira reconstruction of a serially sectioned osmium-ethyl gallate-treated brain.

associated with the visual system, including the reniform body, and the endings of centrifugal neurons from the midbrain.

Serial sections, cut parallel to the rostro-caudal plane of the lateral protocerebrum and stained with reduced silver, suggest the calyces are planar neuropils (*Figure 3B*). But this is deceptive, as shown by sections demonstrating how the deep calycal cytoarchitecture extends rostrally to merge with the base of its column (*Figure 3C*). The total extent and volume of calycal neuropil is surprisingly large as can be seen in brains treated with osmium-ethyl gallate, a technique that selectively colors cell membranes black (*Wigglesworth, 1957*). The method reveals volumes of neuropil in shades of black to light gray, the darker the neuropil, the more densely packed its neuronal processes (*Figures 2A* and *3D,E*; *Figure 6—figure supplement 1B*). The darkness of the calyces reflects the dense packing of the many thousands of intrinsic neurons comprising them. Volumes that are paler are composed of neuronal processes that are stouter and contribute less stainable membrane, as in the overlying gyri.

## Neuronal organization of the calyces (*Figures 5–8*)

Computer-generated reconstructions and rotations of osmium-ethyl gallate stained sections (see Materials and methods and *Figure 3—figure supplement 1*) demonstrate that the calycal volumes

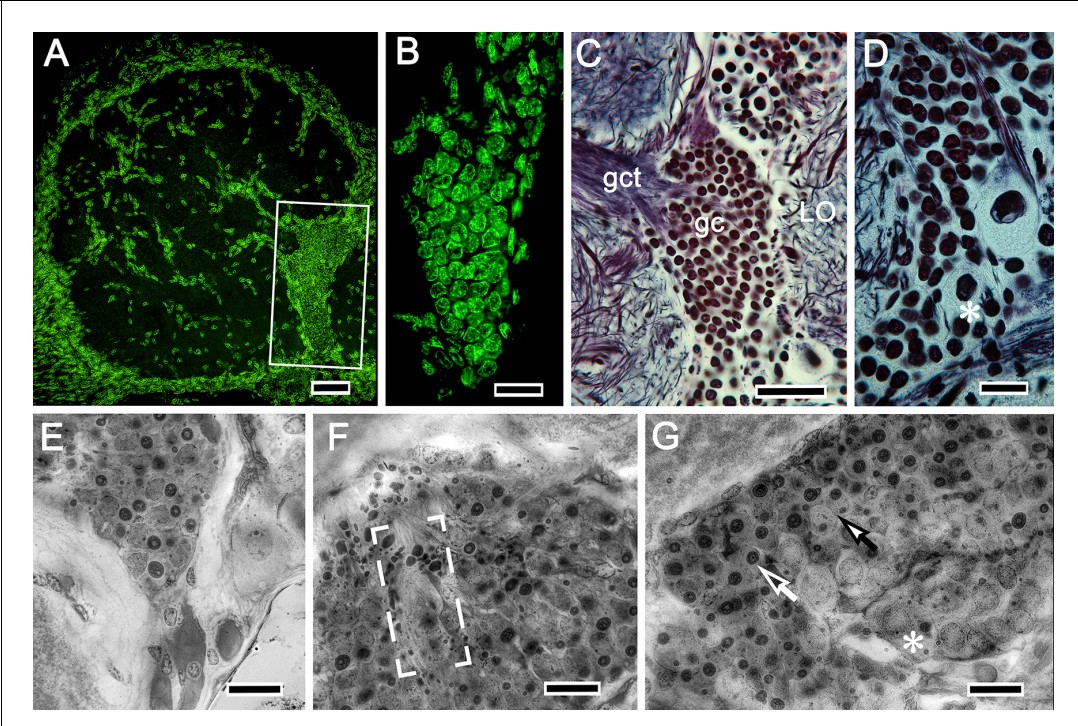

**Figure 4.** Globuli cells supplying intrinsic neurons to the mushroom bodies. In the varunid crab, globuli cells occupy a unique location between the distal margin of the lateral protocerebrum and proximal margin of the optic lobe as shown boxed in **A**. (**B**) As is typical of globuli cells in other taxa (insects, shrimps), they are minute, here measuring less than 6 μm diameter, the next largest being reniform body perikarya (upper right in **C**). (**D–G**) Like other decapods, crabs molt and grow throughout life, their brains also increasing in size. The globuli cell cluster evidences neuroblasts (asterisk in panel **D**), and grouped globuli cells suggest clonal siblings (**E**). Ganglion mother cells are also arranged as aligned groups (**F**) or, as in panel (**G**), dividing (closed arrow) between neuroblasts (asterisk) and fully differentiated perikarya (open arrow). Scale bars, **A**, 50 mm; **B**, 20 μm; **C**, 50 μm; **D-G**, 20 μm.

are topographically elaborate. Each calyx is stratified at deeper levels of the lateral protocerebrum's rostral volume, but, where calycal neuropils extend forwards, their upper borders are defined by irregular arrangements of small peaks and troughs. Two prominent columns, one from each calyx, extend outwards and branch into the overlying gyri (*Figure 3E*). Reduced silver material and Golgi impregnations resolve their elaborate internal organization, which is described in Section 5, below.

Intrinsic neuron processes and terminals of efferent interneurons are the main constituents of the varunid calyces (*Figure 5B*, *Figure 5—figure supplement 1*; *Figure 6B–G*, *Figure 6—figure supplement 1*). They receive inputs from neurons primarily from the deutocerebrum (*Figure 5—figure supplement 1B*), but also from the optic lobes and from its associated neuropils, such as the reniform body (*Figure 6—figure supplement 2*), as also shown for Stomatopoda (*Thoen et al., 2020*). Golgi impregnations and osmium-ethyl gallate staining demonstrate that calyces, across their horizontal and vertical extent, are composed of discrete domains of morphologically distinct networks comparable to the discrete modality-specific zones of Kenyon cell dendrites that define the insect calyces (*Ehmer and Gronenberg, 2002*; *Strausfeld, 2002*).

Most calycal neuropil is characterized by its dense arrangement of small almond-shaped subunits (*Figure 5A*), here referred to as microglomeruli following the term given to comparable elements described from insect mushroom body calyces (*Yasuyama et al., 2002*; *Sjöholm et al., 2006*; *Groh and Rössler, 2011*). However, some peripheral volumes of the calyces comprise more distributed arrangements (*Figure 5A,B*). These appear to receive terminals from axons of relay neurons emerging from fascicles in the eyestalk nerve that are not part of projections from the olfactory lobes (*Figure 3B*).

Discrete fields comprising intrinsic cell processes are recognized by their distinctive and particular morphologies (*Figure 6B–G*, *Figure 6—figure supplement 1*). Each morphological type is defined by the patterning of its initial branches from its globuli cell neurite and the subsequent reticular trajectories made by their tributaries down to the details of their final specializations. The latter indicate points of functional connection and are morphologically comparable to the claws, clasps, and dendritic spines that decorate the dendritic branches of Kenyon cells in *Drosophila* and other insects (*Ito et al., 1998*; *Strausfeld and Li, 1999*; *Strausfeld, 2002*) and the dendrites of homologous intrinsic neurons in the mushroom bodies of marine hermit crabs (*Strausfeld and Sayre, 2020*). As in those taxa, in the varunid calyx these specializations invariably relate to the distribution and patterning of its microglomeruli (*Figure 5*).

Microglomeruli revealed by reduced silver beautifully match the size and mosaic arrangement of microglomeruli resolved by their intense immunoreactivity to antibodies raised against DC0 (*Figure 5*). The bundled branches of intrinsic neurons resolved by antibodies against α-tubulin (*Figure 5D,E*) further resolve microglomeruli as participants of an elaborate organization of orthogonal networks that correspond to distinct morphological types of intrinsic neurons revealed as single cells by Golgi-impregnation (*Figure 6B,C*, *Figure 6—figure supplement 1*). Each microglomerulus signifies the site of an incoming terminal's presynaptic bouton or varicosity (*Figure 5—figure supplement 1*). Combined anti-synapsin immunoreactivity and F-actin labeling (*Figure 5F–G*) further demonstrates the profusion of postsynaptic sites belonging to intrinsic neurons that converge at each microglomerulus. The densely packed presynaptic specializations further confirm the status of a microglomerulus as the site of massive transfer of information from presynaptic terminals onto many postsynaptic channels.

As remarked, the shapes and arrangements of intrinsic neurons and their synaptic specializations contribute to orthogonal networks in the calyces and define calycal domains. However, the domains are not arranged as neatly adjacent territories as in the insect calyx (*Gronenberg, 2001*). Instead, the varunid calyx can reveal domains that extend vertically or horizontally (*Figure 5—figure supplement 1*), or comprise unexpectedly elaborate arrangements that are understood only after they are matched to an appropriately oriented osmium-ethyl gallate section (*Figure 6—figure supplement 1A,B*). One of the most challenging of these arrangements consists of a population of intrinsic neurons defining a cylindrical domain that splays out as it extends toward the proximal margin of the lateral protocerebrum (*Figure 6A*). The entire ensemble comprises densely branched narrow-field arborizations that surround the initial branches of a subset of globuli cell neurites (*Figure 6A,C,D*). The core leads to a terminal expansion that is confluent with the base of one of the two large columns that extends to the overlying gyri. Serial reduced-silver sections reveal this as originating only from calyx 2, indicating that calyx 2 differs from calyx 1 with regard to at least one population of

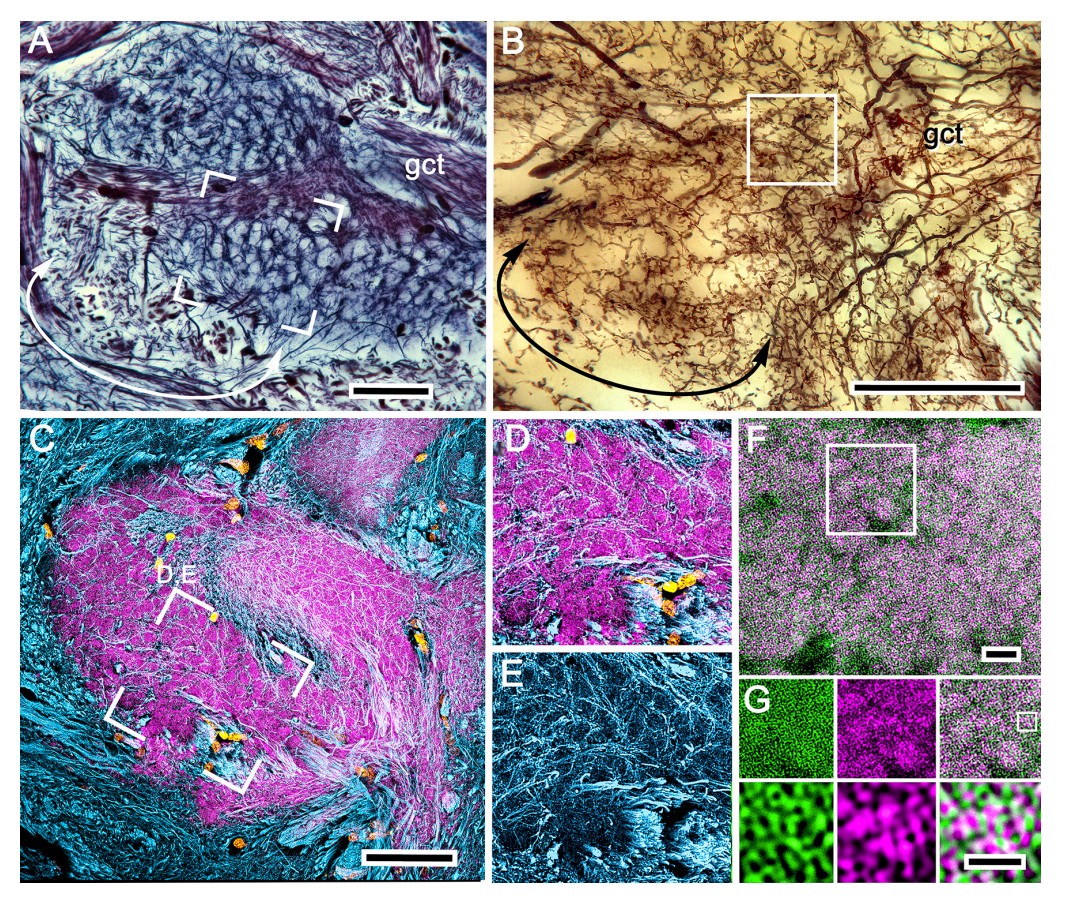

**Figure 5.** Organization of calyx 1: microglomeruli. (**A**) Reduced-silver-stained calyx showing its globuli cell tract (gct) with, at a different level, many of its constituent neurites giving rise to branches that delineate the mosaic organization of microglomeruli. (**B**) Golgi impregnations provide examples of intrinsic neurons, the branches and fine processes of which extend through domains of the calyx that correspond to those revealed both by reduced silver and by antibodies against DC0. A subfield (curved double arrow) in panel **A** corresponds to the denoted branches of the intrinsic neuron in panel **B**. (**C**) Microglomeruli are specifically labeled by anti-DC0 (magenta). Microglomerular size and density in the boxed area corresponds to an equivalent microglomerular organization identified by reduced silver (boxed area in **A**). The mosaic is also reflected by the pattern of all intrinsic neuron processes at that level revealed by anti-α–tubulin (cyan); these enwrap DC0-positive microglomeruli (**D, E**). The same system of microglomeruli is resolved using combined actin/anti-synapsin (**F**). The area identified in a field of Golgi-impregnated intrinsic neuron processes (boxed in **B**) is equivalent to the area of the calyx in panel **F** labeled with actin/anti-synapsin demonstrating the microglomerular mosaic. The boxed area in F is shown in panel **G**. (**G**) The upper row shows actin (green) and synapsin (magenta) separately, with the superimposed images in the right panel. The boxed area in the upper right panel of G indicates a single microglomerulus. This is enlarged in the lower three panels, again showing actin and synapsin as separate images and then combined lower right to demonstrate the density of synapsin-labeled profiles, and thus direct evidence of massive convergence at a calycal microglomerulus. Scale bars, **A–C**, 50 μm; **F**, 10 μm; **G**, 2 μm.

The online version of this article includes the following figure supplement(s) for figure 5:

**Figure supplement 1.** Rectilinear network organization of the calyx and afferent supply.

intrinsic neurons. As in many insect lineages where there are paired calyces, it is such subtle distinctions that demonstrate disparate organization across paired calyces (*Farris and Strausfeld, 2003*). Reduced-silver stains showing axon tracts reveal inputs to calyx 2 originating from a tract that recruits its axons from the reniform body and the medulla. Collaterals from this tract spread into a deep level of calyx 2 neuropil where they spread across microglomeruli bordering one side of the incoming neurites from the globuli cell tract (*Figure 6—figure supplement 2*). The correspondence of this arrangement with insect calyces is considered in the Discussion.

There is one critical exception to the generalization that calycal subdivisions are denoted by different types of intrinsic neurons. The exception is that there is a system of neurons that arborizes uniformly throughout both calyces. These arborizations belong to GAD-immunoreactive local processes

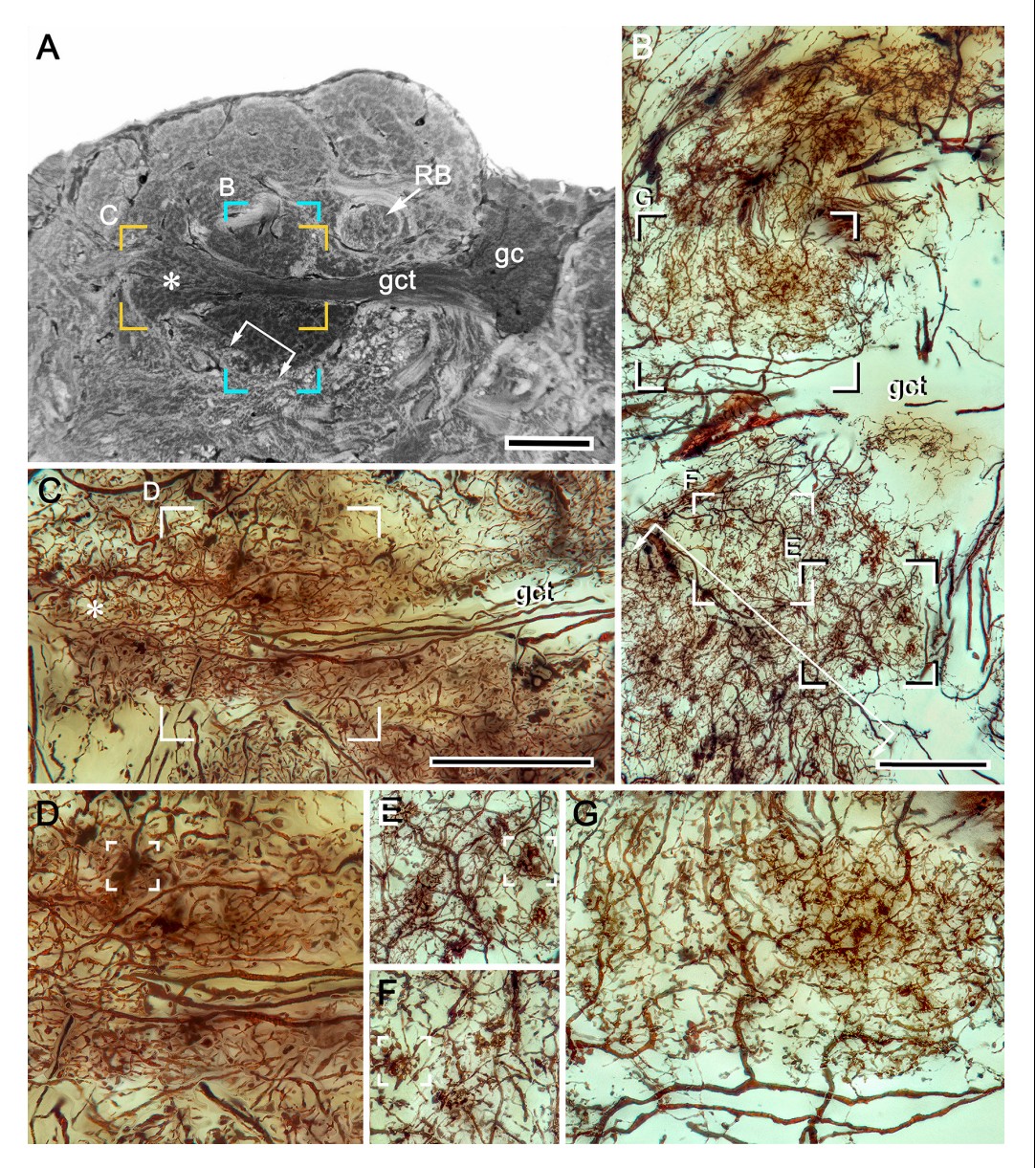

**Figure 6.** Intrinsic cell dendrites define calycal territories. (**A**) Osmium-ethyl gallate resolves calycal territories matching the dendritic fields of morphologically distinct intrinsic neurons. Letters in panel **A** indicate panels showing corresponding locations of Golgi-impregnated dendritic fields shown in (**B** and **C**). Boxed areas in **B** and **C** are enlarged in panels (**D–G**) to demonstrate distinctive dendritic morphologies. The locations of the arrowed bracket and asterisk in **A** correspond to their locations in **B** (the bracket) and **C** (the asterisk). These further indicate the fidelity of calyx organization indicated by these two histological methods. Open boxes in panels **D–F** indicate different morphological specializations of afferent terminals. Abbreviations: RB, reniform body pedestal; gc, globuli cells; gct, globuli cell tract. Scale bars, **A**, **C**, 100 µm, **B**, 50 µm.

The online version of this article includes the following figure supplement(s) for figure 6:

**Figure supplement 1.** Fidelity of intrinsic neuron fields (A) and calycal domains revealed by osmium-ethyl gallate (B).

**Figure supplement 2.** Calyx distinctions.

originating from a dorsal cluster of large cell bodies, each of which provides an anaxonal field that arborizes throughout both calyces, and into each of its domains. (*Figure 7A,B*). These densely packed processes are exceptional in that they do not obey the orthogonal leitmotif of the calyx, as revealed by anti-α-tubulin (*Figure 7C,D*), nor do they reflect the mosaic arrangement of microglomeruli (*Figure 7C–E*). Instead, anti-GAD-immunoreactive branchlets populate all the calycal domains and do so in great numbers, the totality suggesting that local inhibitory elements are distributed

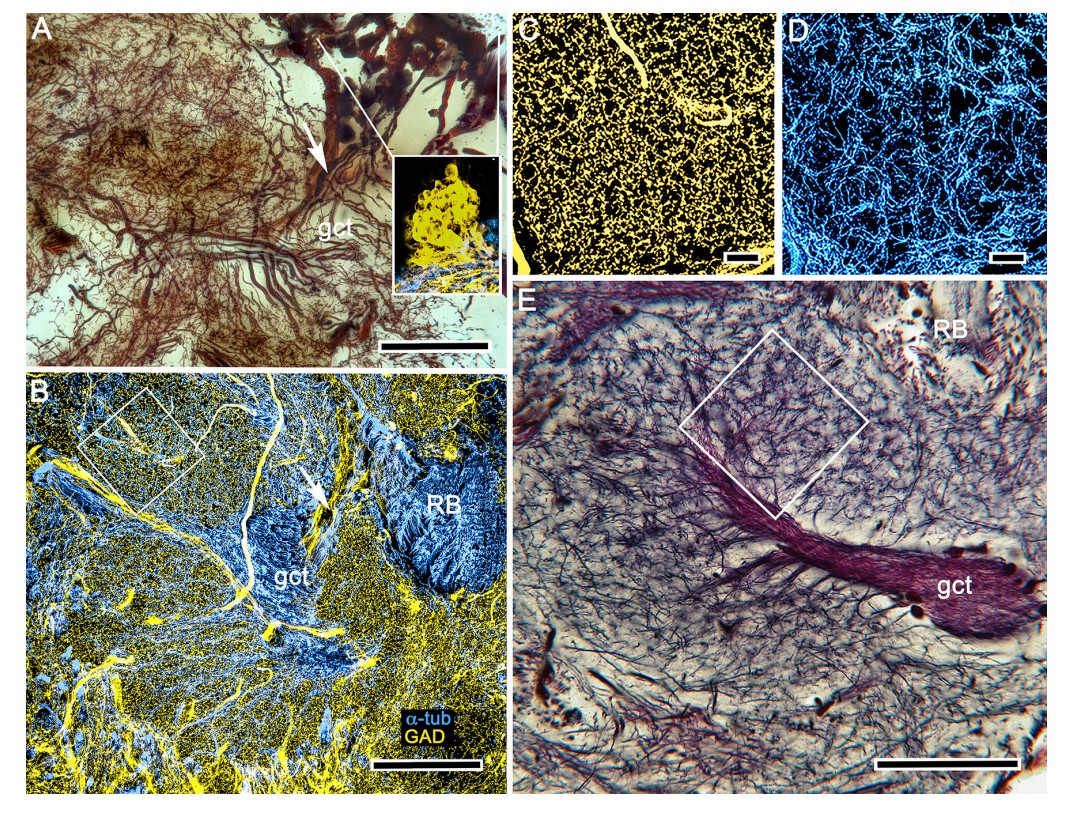

**Figure 7.** Evidence for local inhibition in the calyx by anaxonal neurons. (A) A total of 40–50 large perikarya situated rostrally above the gyri are immunoreactive to glutamic acid decarboxylase (GAD), as shown in the right inset. These neuron cell bodies, which are here Golgi-impregnated, send stout cell body fibers (arrow) to the calyces where they provide widely branching and extremely dense processes that do not follow the patterns of the microglomerular mosaic. (B) GAD immunoreactivity (yellow) reveals their branches and finest processes spreading through every calycal domain and hence positioned to potentially interact with all the intrinsic neuron dendritic fields. (C, D) GAD immunoreactivity in an area shown by the box in panel B, and an equivalent area in panel (E) showing the mosaic of microglomeruli. This mosaic is also evident from the pattern of α-tubulin-immunoreactive (cyan) intrinsic neurons shown in D (same boxed area as in B). Scale bars, A, B, 100 μm, C, D, 10 μm, E, 50 μm.

amongst all intrinsic cell processes irrespective of whether they are in immediate proximity to or distant from a microglomerulus (*Figure 7C–E*).

Whereas GAD immunohistology does not distinguish local calycal domains, these are indicated by the inhomogeneity of fields of 5HT- and TH-immunoreactivity across the calyces (*Figure 8*). The neurons providing these fields also appear to be specific to the calyces and not part of a wider system that extends to other regions of the lateral protocerebrum. Those more distributed systems include densely branched arborizations that populate the optic lobe's lobula and 5HT-immunoreactive neurons that arborize in certain neuropils of the caudal lateral protocerebrum as well as the gyri. These are likewise distinct from the 5HT- and TH-immunoreactive fields that intersect the mushroom body columns, which are described next.

## Transition from the calyces to the columns (*Figures 9* and *10*)

Thus far, we have demonstrated the composition of the calyces as approximately orthogonal networks provided by intrinsic neurons and their arrangements with microglomeruli.

Stomatopod and shrimp mushroom bodies demonstrate a clear transition of neuronal organization between the calyx and its columnar extensions (*Wolff et al., 2017*; *Sayre and Strausfeld, 2019*). In *Hemigrapsus nudus* there is no abrupt transition (*Figure 3*). This ambiguity is seconded in

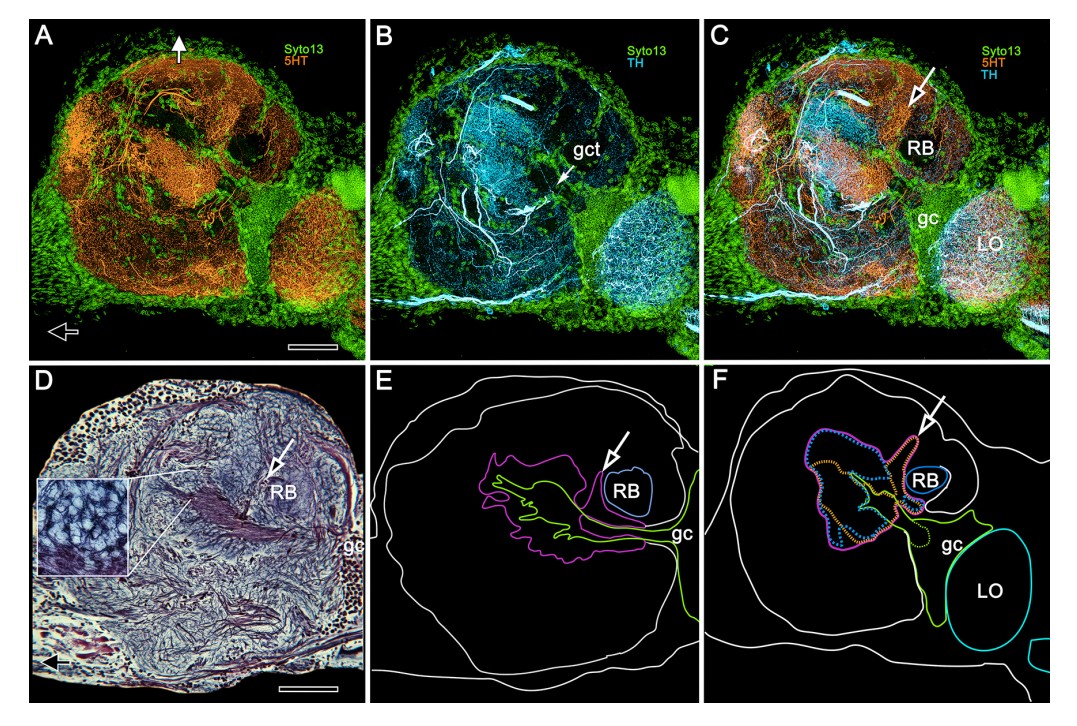

**Figure 8.** 5HT and TH immunoreactivity define different calycal domains. (A,B) Although many volumes of the lateral protocerebrum are distinguished by 5HT (panel **A**, 5-hydroxytryptophan, orange) immunoreactivity, the densest system of branches is in a rostral domain of calyx 1. TH (panel **B**, tyrosine hydroxylase, cyan) defines two domains, one partially overlapping 5HT, the other not (panel **C**). (D) Bodian silver-stained calyx 1 showing its distinctive microglomerular mosaic (inset). (E) Outlined in purple is the total calycal perimeter at this level of sectioning and orientation. The arrow indicates the origin of column 1. (F) Perimeter (purple) of calyx 1 in a section of the rostral lateral protocerebrum slightly rotated around its lateral-medial axis, mapping within it 5HT- and TH-immunoreactivity (as in panels **A–C**) also indicating the origin of column 1 (arrow, as in panel **C**). Abbreviations: RB, reniform body; gc, globuli cell cluster; gct, globuli cell tract; LO, lobula. Scale bars for all panels, 100 μm.

brains treated with osmium-ethyl gallate, which show lateral protocerebral neuropil supplied by globuli cell neurites gradually fading from black to gray, signifying decreasing densities of neuronal membrane, finally giving way to the uniformly pale aspect of the overlying gyri (*Figures 2A* and *3D, E*). 3D reconstructions of the dark volumes show a layered arrangement at deeper layers of the calyces changing to a more corrugated appearance, particularly at its outer surface (*Figure 3—figure supplement 1*). Silver impregnations show that the base and some of the initial length of a column are clearly reticulated (*Figure 9C*). However, these are not microglomeruli. Double Golgi impregnation shows that these networks receive contributions from two sources: the extensions of axon-like processes from intrinsic neuron fields in the calyx; and intrinsic arborizations, many extremely complicated, that originate from thin neurites, presumed to belong to globuli cells (*Figure 9—figure supplement 1*). An additional system of networks extends outwards to flank each calyx's column (*Figure 9B*). These finger-like extensions originate from a tangle of processes situated at a deeper level of the calyces (*Figure 9B*) that does not reflect any relationships with microglomeruli. The likely origin of this system (*Figure 9B*) is not from globuli cells but from a small cluster of cell bodies, some DC0-immunoreactive, situated caudally close to the inner margin of the lobula (*Figure 2C*). We interpret this tangle of processes (*Figure 9B*) as belonging to interstitial local interneurons providing elements to the columns at a level limited to their emergence from the calyces. The topology of these arrangements, including the small column-specific intrinsic arborizations (*Figure 9—figure supplement 1E–K*) is reminiscent of interstitial neurons interposed between the calyces and pedunculus of the honeybee (*Strausfeld, 2002*).

The origins of the columns and smaller extensions from the calyces are further demonstrated by selected levels of silver-stained brains. Serial sections resolve the uneven rostral border of the calyces and the two columns reaching outwards toward the overlying gyri (*Figure 9—figure supplement 2*). The series also demonstrates the agreement between how silver and osmium-ethyl gallate resolve the origins of the columns from the two calyces. These aspects emphasize that the seemingly irregular morphology of the rostral part of the calyces is not irregular by happenstance but is a defining feature of the lateral protocerebrum.

Reduced silver preparations demonstrate that a column is subdivided into parallel components as in insect mushroom bodies, where each subdivision has a distinctive arrangement of intrinsic neurons (*Sjöholm et al., 2005*; *Strausfeld, 2002*; *Tanaka et al., 2008*) and is defined by distinctive patterns of gene expression (*Yang et al., 1995*). What is unexpected is the degree of elaboration observed in the varunid mushroom body. Golgi impregnations demonstrate that, compared with other pancrustaceans, mushroom bodies in a crab are stunningly complex with regard to both their calyces and columns. The varunid's columns consist of parallel intrinsic cell fibers that are almost obscured by numerous other arrangements, the variety of which is compatible with the variety of their morphological fields in the calyces. A silver-stained column originating from calyx 2 (*Figure 9C*) and one revealed by mass Golgi impregnation (*Figure 9D*) both suggest longitudinal subdivisions of the columns. Golgi impregnations resolve discrete and complicated systems of processes, some providing bridgelike connections across them (*Figure 9E*), others having irregular branches, lateral prolongations, and bundled parallel fibers (*Figure 9E,F*). In addition, larger dendrite-like elements arising from stout processes within the columns suggest that those processes may provide outputs to the overlying gyri (*Figure 9D,F*), a feature to which we return in Section 7.

Golgi impregnation has its own peculiar limitations: it is stochastic, revealing probably less than 1% of the total population of neurons, even after double impregnation. Nevertheless, mass impregnation further suggests that the columns from the two calyces are not identical. Although networks of intrinsic cells in both columns receive terminals from branches of the olfactory globular tract, there are clear differences between the columns from calyx 1 and calyx 2. Column 2 is broader and more difficult to discern in reduced silver, in part because its intrinsic processes are extremely thin, many less than a micron in diameter (see section Transitions to gyri below). At first sight, the organization of processes comprising column 2 appears to be chaotic. However, closer inspection reveals subsets of different arrangements of its dendrites and collaterals disposed across its width and at different levels along it (*Figure 9—figure supplement 1A–C*). Reciprocal channels to the column from the gyri are suggested by the presence of descending axons flanking the column, providing varicose collaterals into it (*Figure 9—figure supplement 1A*). These varieties of columnar arrangements are further considered in the Discussion.

Fields of 5HT- and TH-immunoreactive dendrites extend across columns. Here we show fields relating to column 1 from calyx 1 that provide clear resolution of their arrangements from the calycal origin to the interface with gyri (*Figure 10*). Fields are arranged at specific levels along the column, and they intersect different domains across the column, as occurs across the parallel subdivisions of the columns (lobes) of mushroom bodies in *Drosophila* and other insects (*Nässel and Elekes, 1992*; *Liu et al., 2012*; *Aso et al., 2014a*; *Hamanaka et al., 2016*; *Tedjakumala et al., 2017*). The first of these fields occurs at the origin of the column from the calyx (*Figure 10A,B*) above where the neat orthogonal arrangements seen at deeper levels give way to the first appearance of looser arrangements, including parallel fibers (*Figure 10C,F*). Anti-5HT- and anti-TH-immunoreactive fields are clearly segregated (*Figure 10A and D*) at or near the column origin but begin to overlap at more rostral levels closer to the gyri, at locations where osmium-ethyl gallate preparations show the columns branching, with tributaries extending to overlying gyri. At those levels, immunoreactive processes appear to spread laterally into the gyri themselves (*Figure 10G,H*).

## Organization of gyriform neuropils (*Figures 11* and *12*)

Five distinct gyri comprise the outer layer of the rostral lateral protocerebrum (*Figure 11A*). But only four gyri are associated with the two calyces. A fifth gyrus, termed here the proximal gyrus, is situated near the entry of the eyestalk nerve. Its neuronal organization distinguishes it as entirely separate from the mushroom body (*Figure 11A,F*).

Mass impregnations demonstrate at this outer level of the lateral protocerebrum large numbers of efferent neurons and their dendritic processes, many of which originate from clusters of loosely

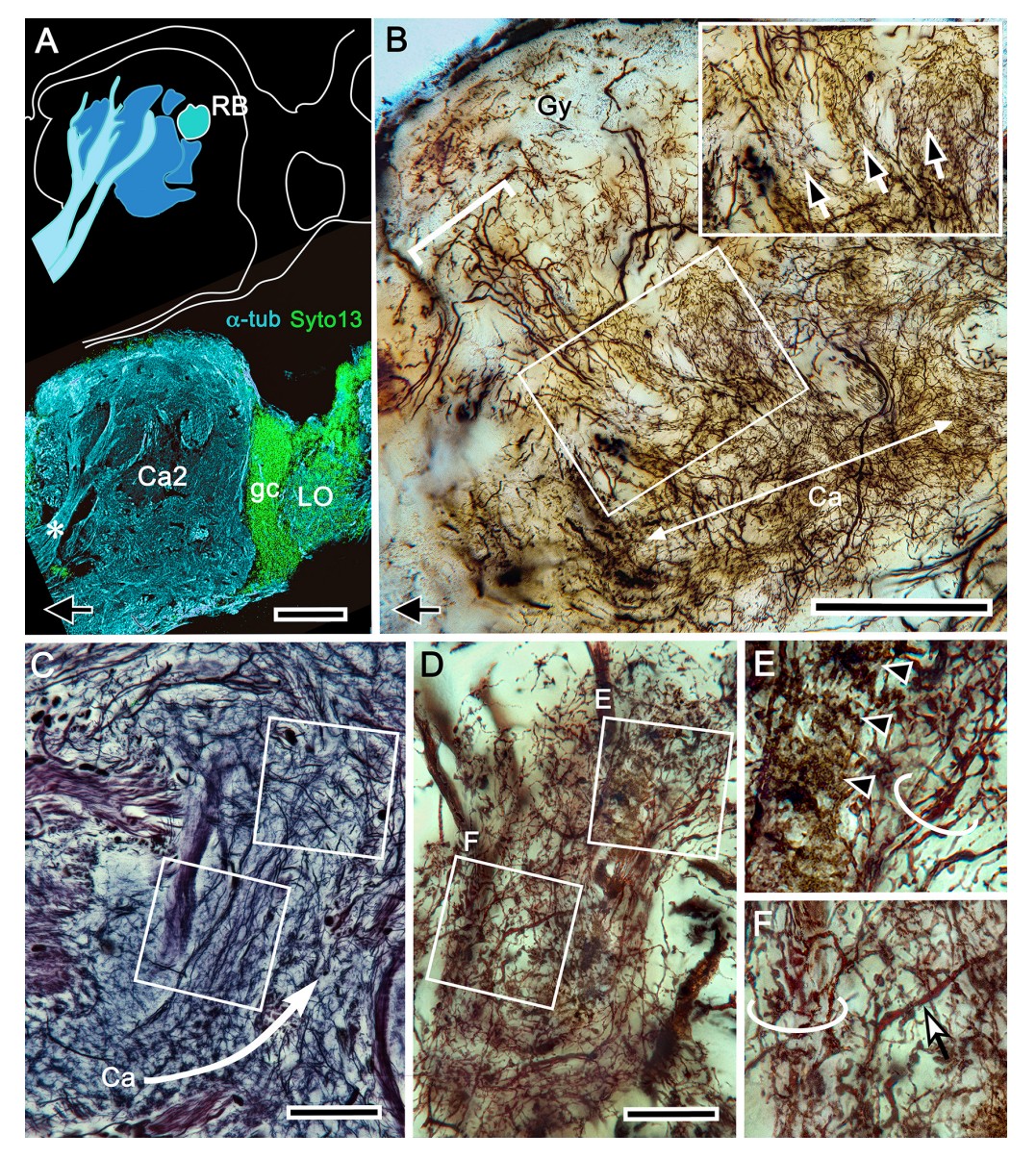

**Figure 9.** Origin of calycal extensions and columns. (**A**) Anti-α—tubulin (cyan) reveals the organization of large fiber bundles and coarse neuropil but, at low magnification, shows neuropils of the calyces as almost featureless. Parts of the calyces are penetrated by incoming tributaries from the eyestalk nerve (asterisk in lower panel). (**B**) A corresponding Golgi-impregnated region demonstrates a system of medium-diameter branched processes that provide fine networks enwrapping the origin of columns. Some networks also extend as finger-like extensions outwards from the calyces. Panel **B** shows one column (bracketed) the base of which is flanked by those arrangements (enlarged in upper right inset indicated by arrows). This extensive system of processes does not derive from the globuli cell clusters, although it is clearly associated with the entire extent of both calyces, suggesting a possible role in calyx-wide modulation. (**C**) As shown by this reduced silver stained section, the column arising from calyx 1 is extremely elaborate, comprising at least two longitudinal divisions each defined by arrangements of intrinsic neuron processes and their branched specializations. The curved arrow in panel C indicates the continuation of intrinsic neuron microglomeruli for a short distance alongside the column. The boxed areas in panel C indicate corresponding levels identified in a Golgi-impregnated column 1 (panel **D**). Boxed areas in D are enlarged in panels (**E**, **F**). In panel **E**, ascending fibers provide short collaterals (arrowheads) across one longitudinal subdivision of the column. Ascending intrinsic processes are arranged as a tangle in two different column subdivisions (circled in panels **E**, **F**). Panel F also shows a large beaded branch (arrowed) of a mushroom body input neuron (MBIN) extending into the column. The arrow, lower left in panels **A**, **B** indicates medial. Abbreviations: RB, reniform body; Ca2, calyx 2; gc, globuli cells; LO, lobula; Gy, gyri. Scale bars, **A**, **B**, 100 µm; **C**, **D**, 50 µm.

The online version of this article includes the following figure supplement(s) for figure 9:

**Figure supplement 1.** Intrinsic neuron organization in columns.

**Figure supplement 2.** The interface between columns and gyri.

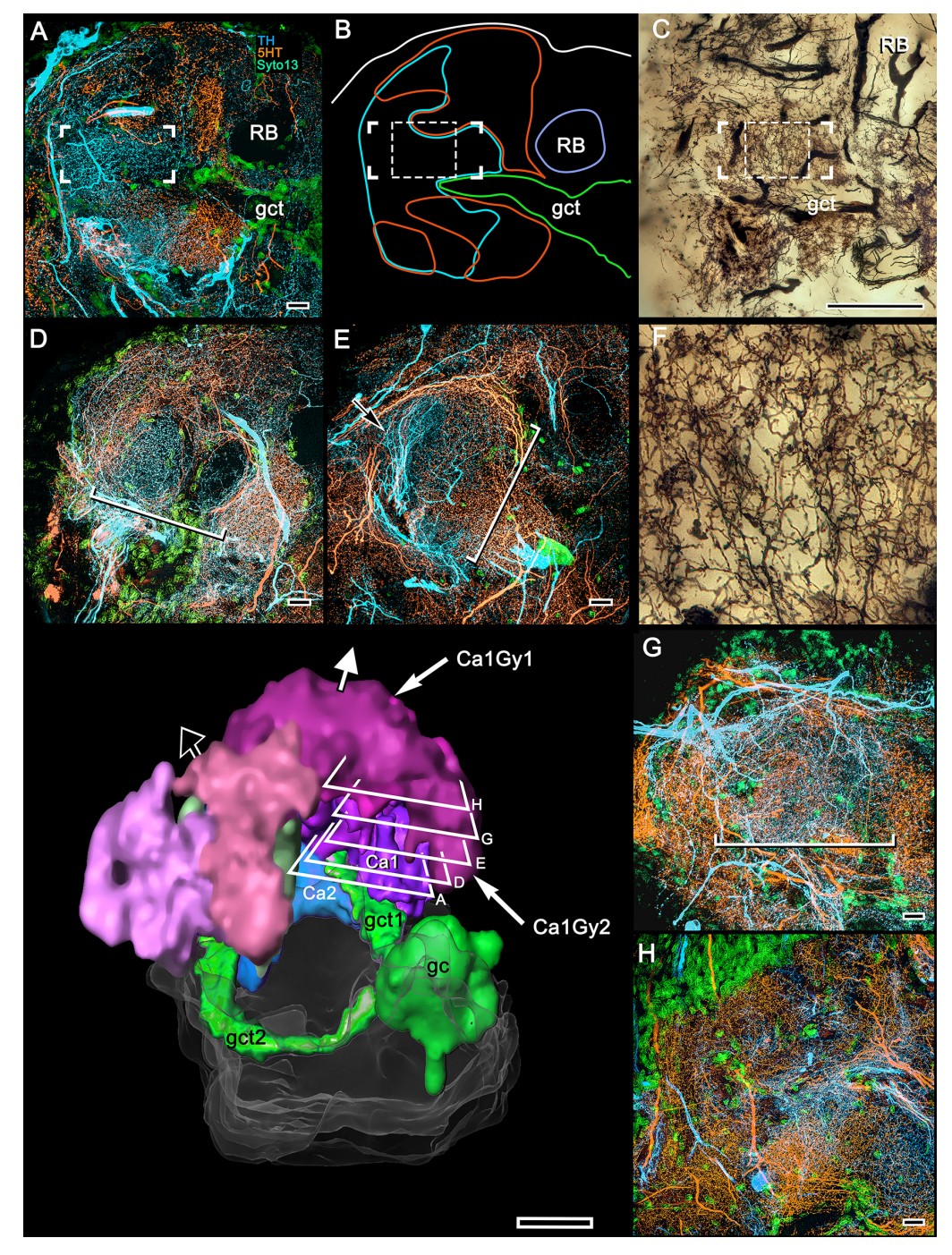

**Figure 10.** 5HT and TH immunoreactivity resolve different levels in the calyx 1 column. Looking downwards through the rostral surface of the LPR, this sequence of confocal slices (**A,D,E,G,H**) starts at the column's origin from calyx 1 (panel **A**) then follows its outward passage rostrally to the level at which the column merges with the overlying gyri (panels **G, H**). (**B**) Shows outlines of 5HT- (orange) and TH-immunoreactive (cyan) fields at level A in relation to the globuli cell tract (gct, green) and the cross section of the reniform body's pedestal (RB). The dendritic field of a TH-immunoreactive cell is corner-bracketed in **A,B** and a smaller portion of it is denoted by a dashed square in **C**. (**C**) Golgi preparation of dendrites in area equivalent to panels **A,B**. Bracketed and dashed areas in (**B,C**) show that at this level, networks of intrinsic neurons have given way to more loosely defined reticulations. (**F**) Enlargement of dashed square in (**C**). At the level of panel **A**, 5HT- and TH-immunoreactive fields are segregated across the origin of the column and continue to be, at least until the level shown in panel (**E**), suggesting 5HT- and TH-immunoreactive fields are restricted to the column's longitudinal subdivisions. TH-immunoreactive fields can be noticeably clustered within a small area of the column's cross section (arrow in **E**) as are MBONs in *Drosophila* mushroom bodies. The width of the column (long brackets in panels **D, E, G**) increases gradually through layers (**D, E,** and **G**)

*Figure 10 continued on next page*

*Figure 10 continued*

(see brackets), and in the gyrus (**H**) its borders are no longer distinguishable. The 3D reconstruction (lower left) shows a view looking into the lateral protocerebrum from its confluence with the optic lobe, its medial axis indicated by the black arrow and the rostral axis by the white arrow pointing upwards. Confocal levels are indicated as well as the two gyri (Ca1Gy1, Ca1Gy2) associated with the Ca1 column and the globuli cell tracts to Ca1 and Ca2 from the globuli cell cluster (gc). Other abbreviations: gct, globuli cell tract; RB, pedestal of the reniform body. Scale bars, **A-E**, **G**, **H**, 10 μm; C and 3D reconstruction, 100 μm.

arranged patches of perikarya that extend over the outer surface of the lateral protocerebrum (*Figure 11—figure supplement 1*). Although stochastically impregnated, the abundance and variety of these neurons suggests the gyri contain dense arrangements of efferent (output) and local interneurons (*Figure 11—figure supplement 1D–I*). Golgi impregnations reveal swathes of axons extending across the gyri, suggesting lateral interactions amongst them. Oblique palisades of dendritic trees, which have branching patterns peculiarly reminiscent of vertebrate pyramidal cells (*Figure 11B*), send axons to fascicles that extend to the midbrain via the eyestalk nerve.

Like deeper levels of the lateral protocerebrum, gyri comprising this superficial level of the lateral protocerebrum are supplied by terminals of afferent neurons, the axons of which radiate out from the eyestalk nerve. Many of these terminals originate from discrete fascicles that comprise the olfactory globular tract, denoting not only their identity as axons of projection neurons from the olfactory lobes but suggesting that different classes of projection neurons comprise different fascicles (*Figure 11D,E*). The terminals of projection neurons are denoted by their swollen boutons, the different morphologies of which are distinctive. For example, terminals spreading out over the surface of gyrus 1, belonging to calyx 1, have rough ball-shaped profiles that are distinct from larger smooth boutons from extremely slender axons supplying the proximal gyrus (pGy in *Figure 11A,B,E*). That gyrus's large allatostatin-immunoreactive processes and the paucity, even absence, of GAD-immunoreactive networks suggests its complete independence from the mushroom body's organization (*Figure 11F*). Nor, as shown by osmium-ethyl gallate, is this gyrus associated with any of the calyces' outward extensions.

In addition to axons extending from the eyestalk nerve providing afferents to superficial levels in the gyri, all levels in the rostral lateral protocerebrum receive an abundant supply of terminals, including many penetrating to deep levels of the calyces (*Figure 11G–K*). Again, it is worth bearing in mind when viewing the images in *Figure 9G,H* that Golgi impregnations likely reveal not more than a few percent of the constituent neurons providing these already dense arrangements of incoming fiber tracts.

## Transitions to gyri

We have shown that four gyri overlie two large columns, each extending from one of the two calyces, as do occasional smaller finger-like extensions. Osmium-ethyl gallate demonstrates that the columns bifurcate close to the gyri (*Figures 2A* and *3E*), a feature also resolved in serial reduced-silver-stained sections (*Figure 9—figure supplement 1*). However, despite the dark staining by osmium that suggests a well-defined transition between columns and gyri, neither silver staining nor Golgi impregnations provide a sharply defined inner margin of a gyrus or an obvious termination of the column. Here we consider neural arrangements that denote the ambiguity of the interface between the mushroom body's columns and their corresponding gyri.

Tangential sections across the gyri demonstrate novel and as yet puzzling aspects of their composition (*Figure 12*). One is the presence of dendritic trees, as in *Figure 12B*, here represented by a neuron, the axon of which joins with others extending to the lateral protocerebral root of the eyestalk nerve. Its dendrites overlap elaborate networks of finely branching processes that terminate in small bead-like swellings suggestive of presynaptic specializations. This network (*Figure 12B*) derives from extremely thin processes that ascend into the gyrus from outward extensions of the calyx (lower bracket in *Figure 12B*). Further examples (*Figure 12C*) show that the dendrites of a gyrus's efferent neurons overlap other arrangements of putative ascending presynaptic fields. In this example, the fields do not appear to be part of an intervening synaptic network, but instead their disposition with the efferent dendritic trees suggests that they likely terminate there (*Figure 12C*). In addition, highly restricted systems of mixed spiny and beaded arborizations suggest local anaxonal neurons that may provide synaptic interfaces between such terminal systems arising from the

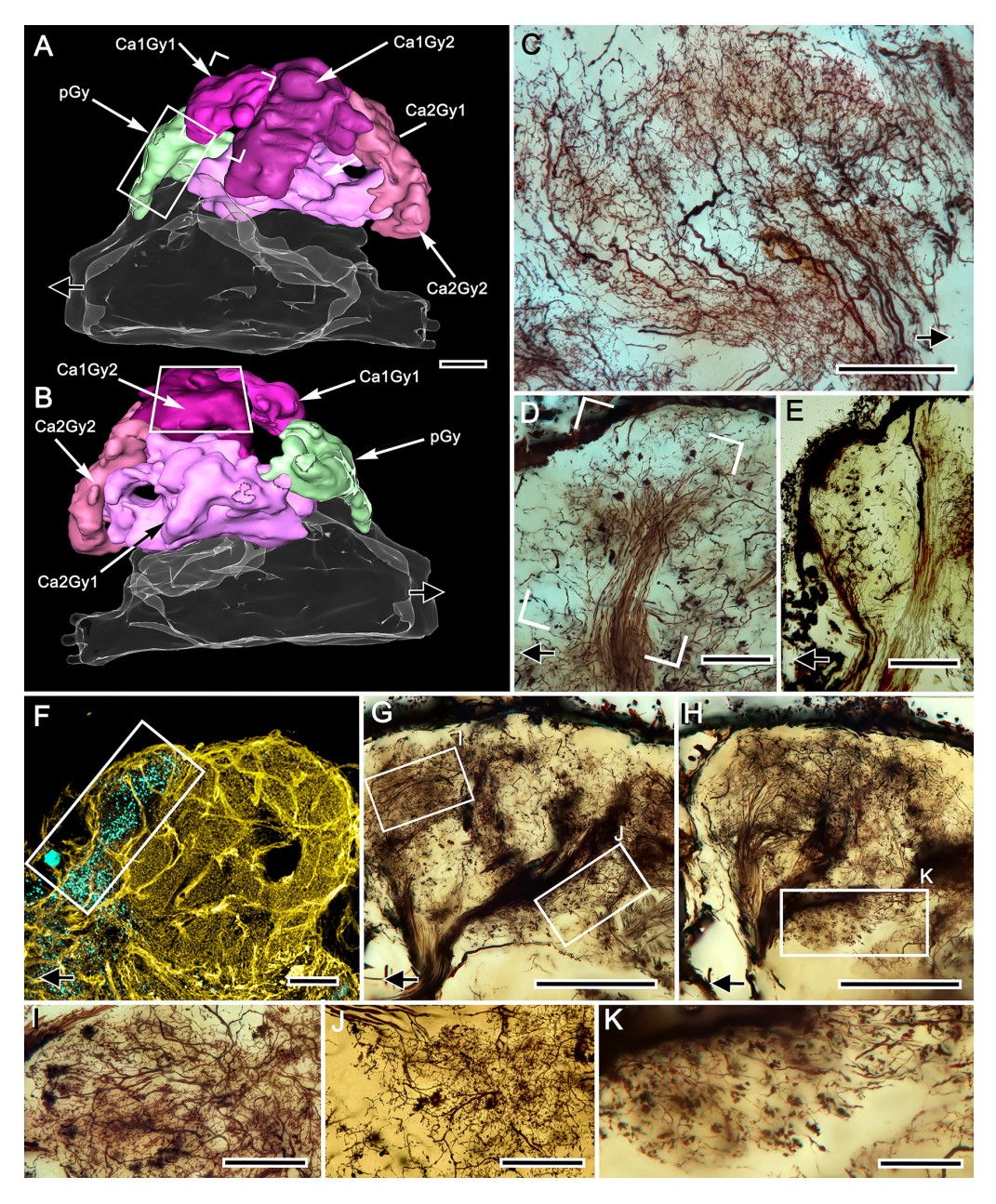

**Figure 11.** Gyriform neuropils: afferent and efferent organization. (A, B). Amira-generated 3D reconstructions of the gyri and their calyces. Black arrows indicate the direction of the eyestalk nerve; likewise in panels C–H. (C) Ensemble of efferent dendrites in gyrus 2 associated with calyx 1 (Ca1Gy2). The represented area is boxed in panel B. (D) Afferent tributary from the olfactory globular tract (olfactory projection neurons) supplying gyrus Ca1Gy1 neuropil (open rectangle in panel A). (E) Smooth oval presynaptic specializations denote the terminals of another olfactory globular tract tributary that exclusively supplies the most proximal gyrus (pGy, rectangle in panel A), which has no obvious connection to the mushroom body. (F) Double-labeling with anti–GAD (yellow) and anti-allatostatin (cyan) of the lateral protocerebrum resolves allatostatin immunoreactivity in the proximal gyrus (rectangle in A and F), which shows little to no GAD immunoreactivity. (G, H) Stacked z-axis projections of Golgi-impregnated tracts, imaged through two successive 50 μm sections showing massive supply by afferent neurons. These reach levels of the lateral protocerebrum that include those at and well beneath the gyriform layer (box I, panel G), including levels of the calyces (box J in panel G and box K in panel H). Notably, many efferent neurons relaying information from gyri (panel H) send their downstream axons from the lateral protocerebrum into these tracts. (I–K) Enlargements from regions indicated in G, H, but at sample thicknesses of less than 20 μm to isolate details obscured in flattened optical sections. Scale bars, A–C, F–H, 100 μm; D,E, 50 μm; I–K, 50 μm.

The online version of this article includes the following figure supplement(s) for figure 11:

**Figure supplement 1.** Perikaryal distributions and gyrus neurons.

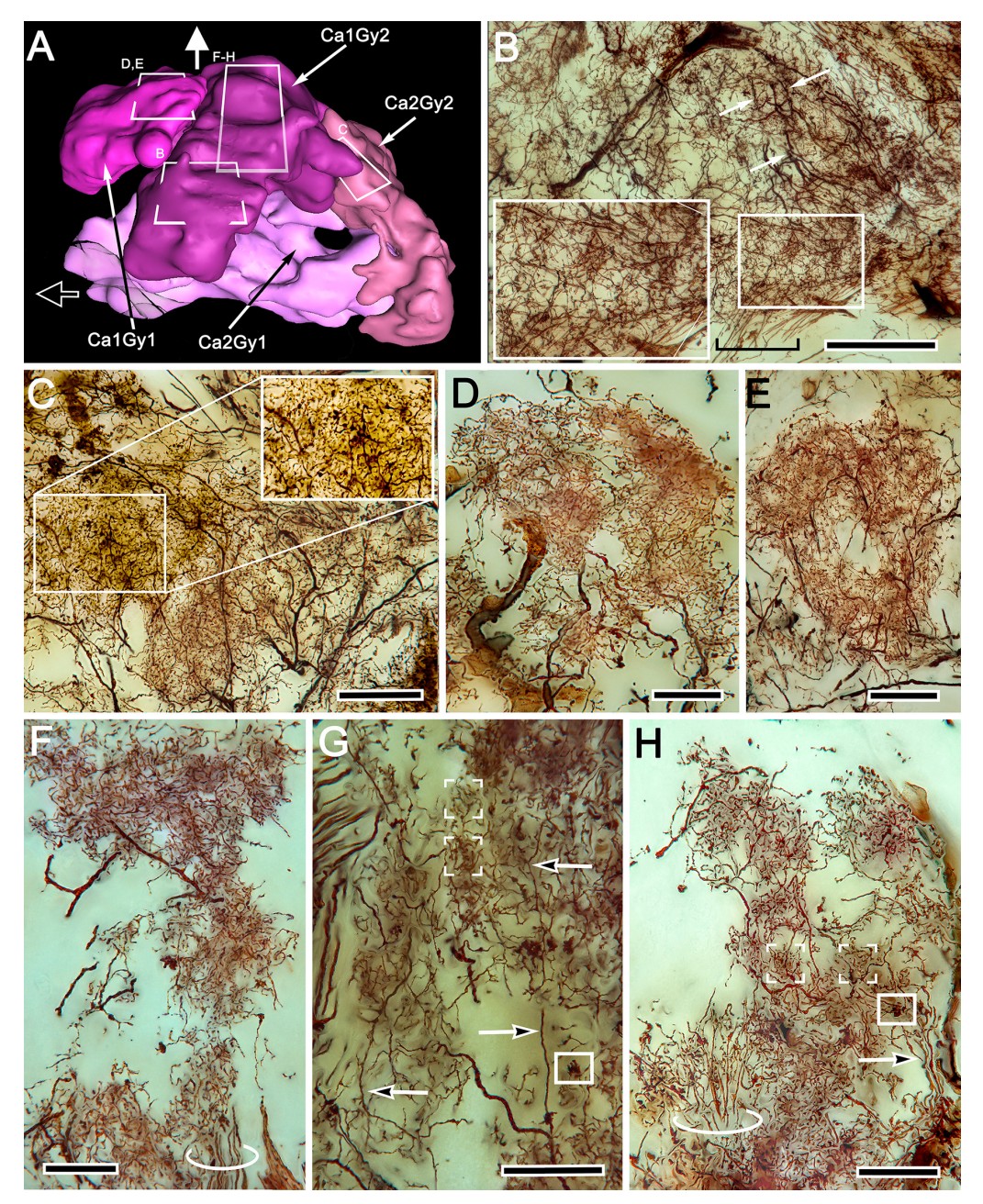

**Figure 12.** Gyrus neuropils and columns: local circuit neurons. (A) 3D reconstruction of gyri associated with calyx 1 and calyx 2, showing approximate locations of networks shown in **B–H**. Open arrow indicates medial, white arrow indicates rostral. (B) Dense network supplied by ascending processes (boxed area is enlarged on left), ascribed to Ca1Gy2 derived from column 1. Dendrites of an efferent neuron (arrows) spread through this layer. (C) Efferent neuron dendrites overlap a broad field of dense terminal specializations. Gnarled bouton-like specializations (boxed and enlarged on right) show that terminals from the eyestalk nerve enter this level. (D–H) Dense and elaborately branched elements interpreted as local interneurons (D, E) and intrinsic elements of the Ca1 column (F–H). Ascending processes (circled in **F,H** and arrowed in **G,H**) are further evidence of local circuits. Dense groups of converging processes (box brackets in **G,H**) correspond in size to afferent boutons (boxed in **G,H**). Scale bars, **B, C**, 100 μm; **D-H**, 50 μm.
The online version of this article includes the following figure supplement(s) for figure 12:

**Figure supplement 1.** Column-gyrus interface organization.

columns and the dendrites of gyral efferent neurons (*Figure 12D,E*). Evidence that arborizations predominantly equipped with presynaptic specializations likely originate from the calyces is indicated by densely packed elements situated distally in the columns. These are equipped with short digiform specializations arranged as small basket-like receptacles matching afferent terminals of unknown origin (*Figure 12G,H*).

The rare Golgi impregnation that resolves one of the columns as a recognizable ensemble of heterogeneous elements within the column's boundaries also demonstrates that its terminus is 'open', meaning contiguous with the gyral neuropil into which some of its fibers project (*Figure 12—figure supplement 1A,B*). Also fortuitous are preparations that demonstrate the terminus of a column coinciding with bundled endings from the olfactory globular tract. Such a preparation (*Figure 12—figure supplement 1B*) also shows that the terminus of the column is open to gyral neuropil around it and that processes extend from the column into the gyrus. Combined labeling with anti-synapsin and actin at the terminus of the column (*Figure 12—figure supplement 1C*) further indicates that sizes and densities of sites indicating synaptic convergence differ systematically across it. The flanking gyral neuropils also show substantial variation in synaptic densities. Inputs to the gyri do not, however, derive solely from calycal columns and afferents from the olfactory globular tract. Reduced silver preparations show that tracts from caudal neuropil associated with the optic lobe extend rostrally to the superficial levels of a gyrus (*Figure 12—figure supplement 1D*), yet another indication of the complexity of organization and its multimodal nature at all levels of the mushroom body and its gyri. The possible significance of this is considered below in the Discussion.

## The reniform body (*Figure 13*)

We have shown that the lateral protocerebrum is divided into a rostral and caudal part (RLPR and CLPR), the former containing the paired mushroom bodies. The mushroom bodies are not, however, the sole components of the RLPR. The gyrus nearest the RLPR's junction with the eyestalk nerve, termed the proximal gyrus (pGy), is distinct and probably to some degree functionally separate from the gyri associated with the mushroom bodies (*Figure 11F*). In addition, another prominent system of circumscribed neuropils is situated near the lateral border of the RLPR, arising from a dorsal group of relatively small perikarya almost adjacent to the optic lobes. These neuropils comprise the reniform body, a center identified in stomatopods by *Bellonci, 1882*, who distinguished it from the very prominent mushroom bodies in that taxon. The same center has been described in detail in the stomatopods *Neogonodactylus oerstedii*, *Gonodactylus smithii*, and *G. chiragra*, and the carideans *Lebbeus groenlandicus* and *Alpheus bellulus*. In all these species, the reniform body has the same morphology: a densely populated bundle of axons, referred to as the pedestal, extends from the dorsal surface of the RLPR to its ventral surface, providing four discrete neuropils called the initial (dorsal), lateral, and the ventrally situated proximal and distal zones (*Wolff et al., 2017*; *Thoen et al., 2020*; *Sayre and Strausfeld, 2019*; *Strausfeld et al., 2020*). In all the cited taxa, the reniform body is situated between the columnar mushroom body and the lobula.

Precisely the same placement is observed in the varunid lateral protocerebrum (*Figure 13*). The reniform body occupies a location lateral to the ensembles of neurons that constitute the mushroom body gyri (*Figures 2* and *13*). Reconstructed from serial osmium-ethyl gallate sections, the reniform body's pedestal lies immediately above, that is, rostral to, the two tracts of neurites from the globuli cell cluster (the globuli cell tract) that provide intrinsic neurons to the mushroom body calyces (*Figure 13C*). In *Hemigrapsus nudus*, the reniform body pedestal extends from a cluster of cell bodies at the dorsal surface of the lateral protocerebrum and extends almost to the lateral protocerebrum's ventral surface, bifurcating into two main tributaries about half way along its length (*Figure 13F*). These give rise to the lateral, distal and proximal zones, each zone composed of substantial volumes of arborizations (*Figure 13D*). The reniform body is thus entirely distinct from the mushroom bodies and is situated distally with respect to their centrally disposed calyces (*Figure 13E*). The reniform body's thick pedestal, albeit columnar in form, is distinct from a mushroom body column in that its constituent axons are naked, lacking spines or other specializations that define a mushroom body. The reniform body's four zones of branching collaterals contribute to structures reminiscent of microglomeruli. However, they are of larger girth, and synapsin-actin labeling shows them to have significantly fewer converging presynaptic elements than a calycal microglomerulus (*Figure 13—figure supplement 1D,E*). Despite these clear differences from the mushroom

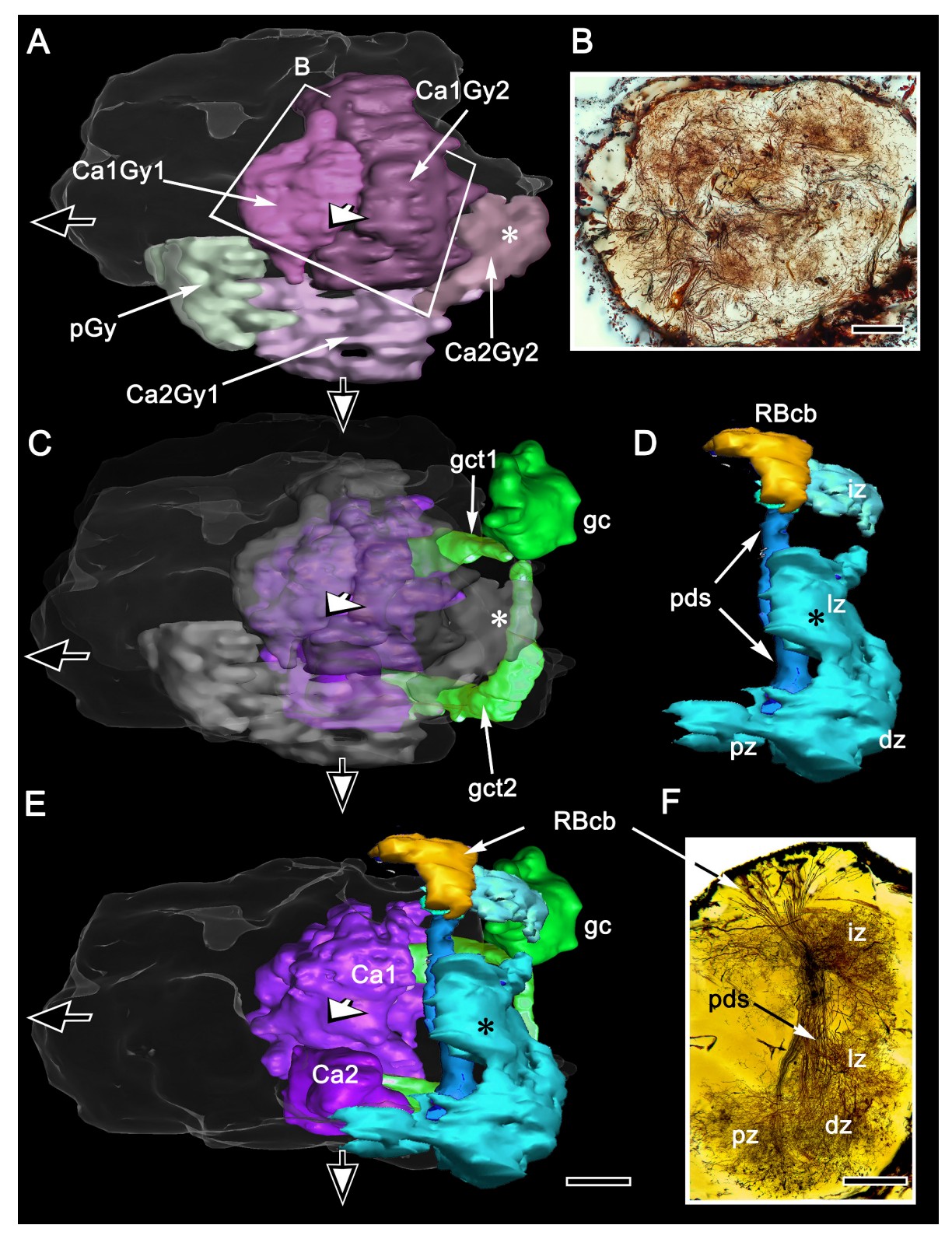

**Figure 13.** Disposition of the reniform body. (**A**) 3D reconstruction of gyri from osmium-ethyl gallate specimens identifies their 2:1 relationship with the paired calyces. These can be compared to the distribution of gyrus neurons resolved by Golgi impregnations (**B** level indicated in A). The Golgi method impregnates a fraction of the constituent neurons, the fields of which are mostly constrained to a single gyrus. Here impregnated neurons are viewed through the lateral protocerebrum's rostral surface. The reconstructions depict the lateral protocerebrum slightly rotated clockwise around the medial-

*Figure 13 continued on next page*

*Figure 13 continued*

lateral axis such that neuropils are viewed from above. In panel **A** (and panels **C**, **E**) the black arrow pointing left indicates medial, the white arrow pointing out of the plane of the page indicates rostral, and the filled arrow pointing downwards ventral. A simplified schematic of the relationships between the calyces and their cognate gyri is provided in *Figure 13—figure supplement 2*. (**C**) The globuli cell cluster (gc) provides two globuli cell tracts (gct1, gct2) to the two mushroom body calyces Ca1, Ca2. The lateral lobe of gyrus 2 associated with calyx 2 (Ca2Gy2, indicated by the asterisk in panel **A**) reaches out distally and partially fills a gap (asterisk in **C**) between two processes (the lateral and dorsal zones, iz and lz) originating from the reniform body's pedestal (pds). (**D**) The reniform body pedestal (blue) is a dense fiber bundle originating from a dorsal cluster of small cell bodies (RBcb, yellow) at the dorsal surface of the rostral lateral protocerebrum. The reniform body's neuropil volumes are colored cyan. The pedestal's initial and lateral zones (iz, lz) are separated from its ventrally dividing branches, shown Golgi-impregnated in panel (**F**), that provide the distal and proximal zone neuropils (dz, pz). (**E**) Combined volumes from panels **A**, **C**, and **D** demonstrating the position and modest dimensions of the reniform body relative to the mushroom bodies, comprising the calyces (Ca1, Ca2), columns and associated gyri. Scale bars, **A-C**, **B**, **F**, 100 μm.

The online version of this article includes the following figure supplement(s) for figure 13:

**Figure supplement 1.** Comparison of synaptic densities associated with the calyces and reniform body.
**Figure supplement 2.** Calyx-gyri topographical relationships.

body, the reniform body is strongly immunoreactive to anti-DC0, showing that it too is a learning and memory center, as proposed by *Maza et al., 2016*.

## Discussion

### Unique organization of the varunid mushroom body

Studies by Hanström in the 1920s and 1930s on the brains of malacostracan crustaceans concluded that mushroom bodies equipped with calyces and columns (lobes), as they are in insects and mantis shrimps (*Bellonci, 1882*), have been reduced in decapods to columnless centers lying immediately beneath the rostral surface of the lateral protocerebrum. Recent comparisons of the brains of reptantian decapods have confirmed that its lineages share the evolutionary trend of reduction and loss of the mushroom body column (*Sayre and Strausfeld, 2019*). Those mushroom bodies nevertheless retain most elements of ancestral networks: diffusely arranged in Axiidea, Astacidea, and Achelata (*Sayre and Strausfeld, 2019*), but precisely stratified in terrestrial and marine Anomura (hermit crabs: *Harzsch and Hansson, 2008*; *Wolff et al., 2012*; *Sayre and Strausfeld, 2019*).

Ocypodidae (fiddler crabs) and Varunidae (shore crabs), two lineages of Brachyura (true crabs), do not conform to this trend. As demonstrated previously and again here, large DC0-positive domains cover much of the varunid lateral protocerebrum, suggesting relatively large (for an arthropod) learning and memory neuropils (*Strausfeld et al., 2020*). The proposition that the reniform body is the crab's mushroom body (*Maza et al., 2016*; *Maza et al., 2021*) is refuted by Golgi impregnations and 3D reconstructions of the lateral protocerebrum demonstrating the reniform body as entirely distinct from the huge DC0-positive mushroom body adjacent to it (*Figure 13*). Similarity, conjunction, and congruence are established criteria for assessing phenotypic homology (*Patterson, 1988*). In all the lineages where reniform bodies have been identified these criteria exclude it as mushroom body homologue.

That the brachyuran mushroom body has until now remained unrecognized can be attributed to its unexpected and radical reorganization. Even its population of conservatively estimated 22,000 densely packed globuli cells is not where it would be expected to be. Based on observations of other pancrustaceans, the expectation is that globuli cells are clustered over the rostral surface of the lateral protocerebrum, close to where it joins the eyestalk nerve (see *Sayre and Strausfeld, 2019*). But in *Hemigrapsus nudus* the globuli cell cluster is sandwiched between the lobula and the lateral protocerebrum. As in the shore crab *Carcinus maenas*, where new neurons are generated lifelong in the antennular lobes and lateral protocerebrum (*Schmidt, 1997*; *Hansen and Schmidt, 2001*), osmium-ethyl gallate histology on *H. nudus* indicates that new globuli cells also continue to proliferate, as do neurons supplying the reniform body (*Maza et al., 2016*). In addition to the unique disposition of the globuli cells, the reorganization has inverted the entire mushroom body such that its two voluminous calyces are deeply buried in the rostral domain of the lateral protocerebrum

where they receive inputs from the deutocerebrum's olfactory lobes and the protocerebrum's optic lobes.

We show here that each calyx gives rise to a large column extending outwards and merging with broad overlying gyri that form the rostral surface of the lateral protocerebrum. The gyri are pillowed into folds and sulci thus forming a cortex-like architecture. This system receives additional afferent supply and gives rise to ensembles of pyramidal cell-like efferent neurons, the axons of which extend

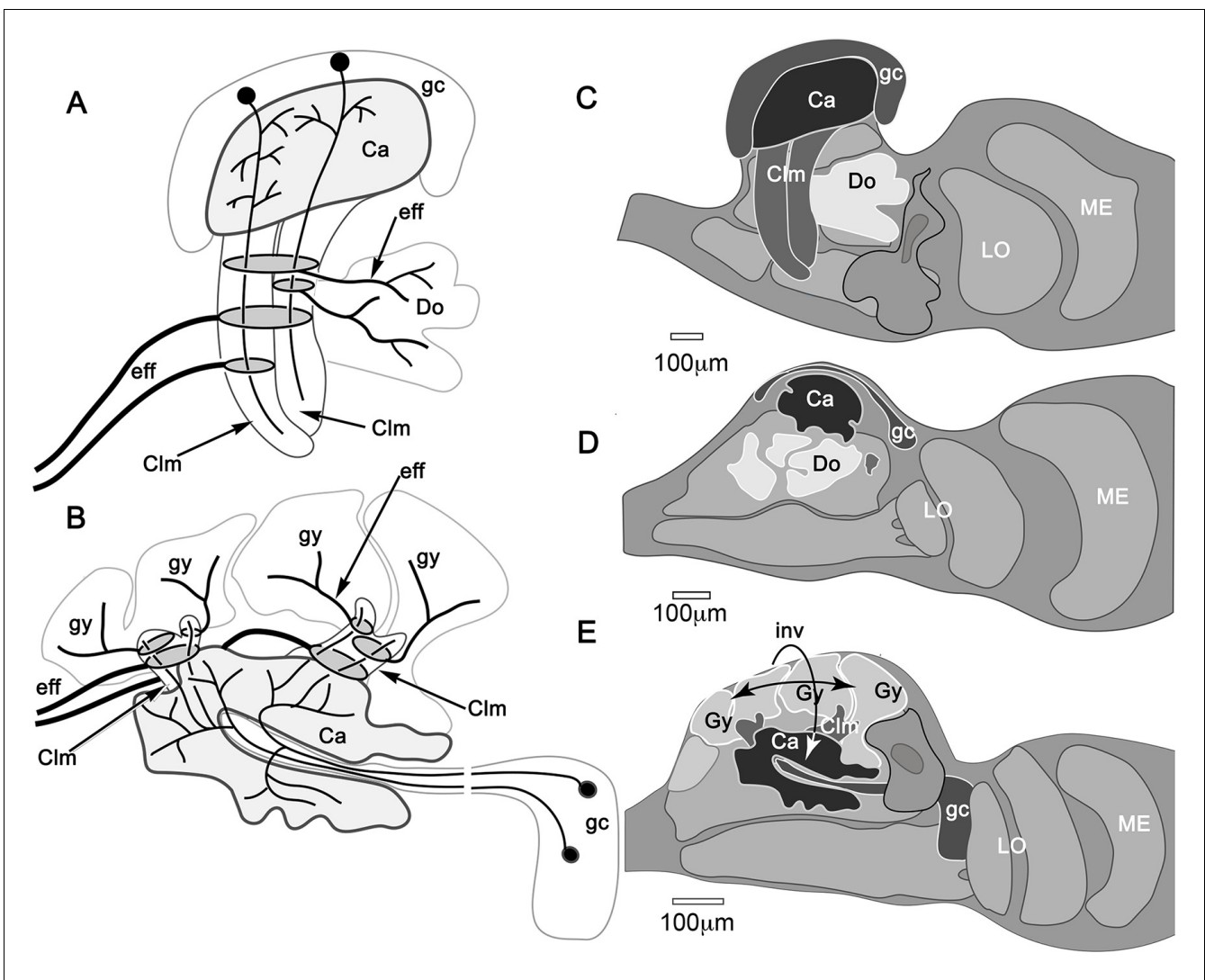

**Figure 14.** Mushroom body inversion and gyriform organization. (**A, B**) Schematics of corresponding neuron ground patterns. (**A**) Morphological traits defining the mushroom body are consistent despite inversion. In Stomatopoda, intrinsic neuron dendrites corresponding to the Kenyon cells of insect mushroom bodies occupy specific domains (here levels) in the calyces (Ca). Their prolongations down the columns are intersected by efferent dendritic fields (shaded ovals) the axons of which (eff) extend medially (left) to the midbrain (not shown) or to domains (Do) deeper in the lateral protocerebrum. (**B**) In *Hemigrapsus nudus*, the mushroom body is inverted. Intrinsic neuron dendrites occupy characteristic planar territories in the calyces (Ca), with rostrally projecting extensions in the columns intersected by efferent neuron dendrites. As in stomatopods, axons of some efferent neurons (eff) extend medially to the superior protocerebrum of the midbrain (not shown). Other efferents supply the gyriform neuropils overlying the lateral protocerebrum (Gy). (**C–E**) Comparisons of mushroom bodies in Malacostraca. (**C**) In the stomatopod *N. oerstedii*, globuli cells cover the calyx; columns project caudally into the lateral protocerebrum neuropil. Neuropil domains (Do) within the lateral protocerebrum receive inputs from the lobes. (**D**) In the crayfish *Procambarus clarkii,* the lobeless mushroom body lies at the lateral protocerebrum's rostral surface. Neuropils (Do) in the lateral protocerebrum receive its outputs. (**E**) In *H. nudus*, calyces are deeply buried in the lateral protocerebrum. Columns extend rostrally to an overlying gyriform neuropil, the entire arrangement indicating mushroom body inversion and expansion of the rostral gyri. Abbreviations: Ca, calyx; Clm, mushroom body column; Do, domains receiving mushroom body efferents; eff, efferent neurons; gc, globuli cells; Gy, gyri; LO, lobula; ME, medulla.

to the eyestalk nerve. Throughout this massive structure, the mushroom body and its gyri are strongly immunoreactive to antibodies raised against DC0.

Although the crab mushroom body appears vastly different from the right-way-up 'traditional' insect or stomatopod homologue (*Figure 14*), it nevertheless shares their defining traits. For example, intrinsic neurons originating from globuli cells contribute to the brachyuran calyces and columns, as do the Kenyon cells of an insect mushroom body (*Figure 14A,B*). As in stomatopods, shrimps, and insects, the axon-like processes of intrinsic cells are clustered in their respective fascicles, as they are in any one of the longitudinal divisions of the insect mushroom body columns, where short intermingling collaterals can provide orthogonal connectivity (*Sjöholm et al., 2005*; *Strausfeld et al., 2003*; *Tanaka et al., 2008*). Other longitudinal divisions of the varunid mushroom body column are populated by intrinsic processes that interweave to provide rectilinearity, as they do in one of the two mushroom body columns of the shrimp *Lebbeus groenlandicus* (*Sayre and Strausfeld, 2019*), or in the stratified dome-like mushroom body of hermit crabs (*Wolff and Strausfeld, 2015*). We show here that true crabs express the whole panoply of intrinsic cell arrangements. These include segregated and intermingling 'parallel fibers' as in Stomatopoda or the core lobe (column) of *Drosophila* (*Wolff et al., 2017*; *Strausfeld et al., 2003*); rectilinear networks that extend part way into a column from the calyx; dense arrangements of intrinsic processes extending across the columns comparable to certain neurons populating the mushroom body column in honey bees or crickets (*Strausfeld, 2002*; *Hamanaka and Mizunami, 2019*); and arrangements comparable to the longitudinal laminations of the mushroom body lobes (columns) of Dictyoptera and Lepidoptera (*Sinakevitch et al., 2001*; *Sjöholm et al., 2005*). In short, the crab mushroom body column appears to embody all known arrangements across Pancrustacea. If each arrangement of a specific type of intrinsic neurons represents a distinct computational property, as suggested from studies of honeybees and fruit flies (*Perisse et al., 2013*; *Traniello et al., 2019*), then the crab would appear to be singularly well-equipped. The same applies to its calycal arrangements, which in *H. nudus* are elaborate and discretely partitioned into territories denoted by different morphologies of intrinsic neurons and afferents. This zonal organization corresponds to the more compact and highly ordered modality-specific domains typifying hymenopteran calyces (*Gronenberg, 2001*), and, to a lesser extent, *Drosophila* calyces as well (*Yagi et al., 2016*; *Li et al., 2020a*).

The varunid lateral protocerebrum has two calyces, each providing a massive column. This doubling of the calyx has many precedents. In the stomatopod lateral protocerebrum there are four adjacent columns, each associated with a distinct ensemble of globuli cells and hence a calyx. That organization has been compared to the *Drosophila* mushroom body, which is a composite neuropil of four hemi-mushroom bodies derived from four clonal lineages, as it is in the honey bee (*Ito et al., 1997*; *Farris et al., 2004*; *Wolff et al., 2017*). Consequently, the mushroom bodies in an insect's lateral protocerebrum are paired but completely fused. The varunid has two calyces in immediate contact, but not quite fused as no intrinsic neuron has been seen that extends its field of processes into both. Only one calyx (Ca2) appears to receive input from a tract of axons carrying relays from the medulla and from the reniform body. Such an asymmetric input to twin calyces is not unique: a comparable arrangement is found in the *Drosophila* mushroom body, where visual inputs terminate in an 'accessory calyx' attached to one side of the fused calyx (*Vogt et al., 2016*; *Li et al., 2020b*). A similar arrangement involving visual inputs has been described from the swallowtail butterfly (*Kinoshita et al., 2015*).

Of particular note is the extensive organization of GAD-immunoreactive arborizations that extend into every domain of the *H. nudus* calyces. These provide dense systems of minute processes, the dispositions of which would allow synaptic contact with every neuronal process supplying a microglomerulus. Accepting that varunid intrinsic neurons correspond to insect Kenyon cells, these GAD-immunoreactive systems suggest correspondence with wide-field anaxonal inhibitory neurons that extend throughout the *Drosophila* calyx ('APL neurons': *Liu and Davis, 2009*; *Amin et al., 2020*). Consequently, such putatively inhibitory anaxonal neurons in the crab suggest a global role in local synaptic inhibition at microglomeruli. Arrangements of aminergic neurons in the calyces and columns further correspond to mushroom body input and output neurons recognized in *Drosophila* (*Aso et al., 2014b*; *Owald et al., 2015*). It is also notable that highly localized and densely branching TH- and 5HT-immunoreactive neurons occur at different levels of the crab's calyces. These correspond to similar dispositions of immunoreactive neurons in the layered calyces of columnar mushroom bodies, including those of stomatopods and shrimps (*Wolff et al., 2017*; *Sayre and*

*Strausfeld, 2019*). Neurons in the *Hemigrapsus* mushroom body having the same immunoreactive signatures are associated with columns, from their origin at the calyces out to the level of their 'fuzzy' interface with gyri and also in the gyri. Their successive levels in a column approximate the arrangements of output neurons from successive partitions of insect mushroom bodies, as exemplified in cockroaches and *Drosophila* (*Li and Strausfeld, 1997*; *Tanaka et al., 2008*; *Aso et al., 2014b*); and in crustaceans by the successive fields of efferent neurons in the mushroom bodies of Stomatopoda, Stenopodidea, and Caridea (*Wolff et al., 2017*; *Sayre and Strausfeld, 2019*; *Strausfeld et al., 2020*).

## Is the organization of gyri an outlier, and what is its homologue in other lineages?

Comparisons of the varunid brain with those of other pancrustaceans meet an impasse when considering the lateral protocerebrum's system of gyri, a cortex-like feature that appears to be special to Brachyura. As described above, there is no obvious terminus of a mushroom body column: its efferent neurons merge with systems of local interneurons in the gyri and, with these local interneurons, they extend amongst the dendrites of efferent neurons that leave the gyri. If these arrangements do not correspond to organization in other pancrustaceans and mandibulates, then what developmental events might have given rise to such a departure from the reptantian ground pattern?

Three developmental studies on the malacostracan lateral protocerebrum should be noted. Two demonstrate early stages of the formation of the larval lateral protocerebrum, in which there is not yet evidence for morphogenetic reorganization in what will become the lateral protocerebrum (*Harzsch and Dawirs, 1996*; *Sintoni et al., 2007*). The third demonstrates that massive reorganization and new generation of neuropils can accompany late development in the pelagic larva (*Lin and Cronin, 2018*). All of those authors have indicated that crustaceans can undergo radical metamorphosis during the change from a pelagic larva to a preadult, the larval stage providing pools of neural precursor cells for subsequent development. Those precursors would likely contribute not only to the formation and differentiation of adult centers in the lateral protocerebrum but also to any taxon-specific morphogenic rearrangements.

Might the relationship of the insect mushroom body with the rest of the brain suggest an origin for the gyri? In insects, many of the mushroom body efferent neurons terminate in neuropils of the superior medial protocerebrum (honeybee: *Strausfeld, 2002*; cockroach: *Li and Strausfeld, 1997*, *Li and Strausfeld, 1999*; *Drosophila*: *Ito et al., 1998*; *Tanaka et al., 2008*; *Aso et al., 2014a*; *Aso et al., 2014b*). This is the most rostral part of the forebrain in insects, comprising dense networks of local interneurons that are interposed between the terminals of mushroom body outputs and the dendrites of interneurons whose axons terminate in successive strata of the central complex's fan-shaped body (*Phillips-Portillo and Strausfeld, 2012*). A comparable arrangement in crustaceans with eyestalks requires that efferent neurons (particularly TH-immunoreactive MBONs) from the mushroom bodies extend axons from that distant origin into the midbrain's superior protocerebrum. These connections are suggested by silver-stained brains of *H. nudus* showing numerous axons from the eyestalk nerve directly targeting the medial protocerebrum. This speaks against the gyri being medial neuropils displaced distally.

One possible interpretation of the gyri is that they represent enormously expanded versions of tubercles, swollen ends of a column typifying mushroom bodies of certain caridid shrimps (in Stenopodidae and Thoridae: *Sayre and Strausfeld, 2019*) and insects (Zygentoma, Ephemeroptera: *Farris, 2005*). Tubercles of caridid (or insect) mushroom bodies are compact, comprising tangled extensions of intrinsic neuron processes associated with a concentrated system of TH- and 5HT-immunoreactive processes intersecting that level of the column. Those arrangements are absent from gyri. Although there are TH- and 5HT-immunoreactive fields in gyri, these do not appear to interact with intrinsic fibers in the columns. Nevertheless, that gyri are as strongly DC0-immunoreactive as the calyces and columns suggests that they play crucial roles in memory acquisition and, or, maintenance. There is one gyrus (pGy), however, that is not obviously DC0-immunoreactive and is quite distinct from the others (*Figure 11A,B,F*). This gyrus is situated most proximally, near the entry of the olfactory globular tract into the lateral protocerebrum from the eye stalk nerve. It is denoted by its association with allatostatin-positive processes. Its afferent supply suggests a role in olfaction, but its sensory associations and roles in behavior await functional studies.

Far more elaborated than a tubercle, gyri comprise many morphologically distinct types of neurons, including efferents of unknown antigenicity. The presence in gyri of local interneurons and rectilinear networks further indicates an identity that is indirectly associated with the mushroom body. A clue is suggested by other Reptantia, in which certain mushroom body outputs target neuropils at deep levels of the lateral protocerebrum (*Mellon et al., 1992a*; *Mellon et al., 1992b*; *McKinzie et al., 2003*). If these less well-studied neuropils became inverted, they would overlie the inverted brachyuran mushroom bodies as do the gyri (*Figure 14B,E*). In that scenario, the relationship between a mushroom body and its target neuropils is essentially the same as in other reptantians (*Figure 14D*), except a consequence of inversion is that those neuropils have been able to undergo unparalleled lateral expansion and gyrification.

## The varunid mushroom body in the context of pancrustacean evolution and cognition

The organization of traits defining mushroom bodies is phenotypically identical in stomatopods and insects (*Wolff et al., 2017*). The same characters have since resolved divergent homologues in all malacostracan crustaceans except Brachyura, the most recent lineage of decapod crustaceans (*Wolfe et al., 2019*). By identifying and describing the mushroom body of the shore crab, the present study completes the survey of mushroom body diversification and phenotypic correspondences, thereby enabling future transcriptomic verification or rejection.

In considering the possible significance of the varunid mushroom body being larger and more elaborate than that of any other pancrustacean, it is relevant to consider a well-studied proxy that shares with mushroom bodies circuit and genetic correspondences, as well as cognitive properties. For example, it has been suggested that the cerebellum, caudal to the vertebrate midbrain-hindbrain boundary, would be a fitting proxy because of comparable arrangements of parallel fibers (*Schürmann, 1987*; *Farris, 2011*; *Li et al., 2020a*). However, mushroom bodies are situated rostral to the arthropod brain's deutocerebral-tritocerebral boundary. A homologous location is occupied by the hippocampi, rostral to the vertebrate midbrain-hindbrain boundary (*Bridi et al., 2020*). Mushroom bodies and hippocampi are restricted, respectively, to the protocerebrum and its vertebrate homologue the telencephalon (*Hirth and Reichert, 1999*). Neuroanatomical arrangements that are intensely immunoreactive to antibodies against DC0 are shared by the hippocampus and mushroom bodies. At least 16 orthologous genes are required for the same functions in both (*Wolff and Strausfeld, 2016*). Gene expression profiling of early mushroom body development in *Drosophila* and of the murine pallium resolve further correspondences (*Tomer et al., 2010*).

A range of behaviors supported by both the insect mushroom bodies and hippocampus relate to allocentric memory, recall of place, and the use of space (*Krebs et al., 1989*; *López et al., 2003*; *Salas et al., 2003*; *Ladage et al., 2009*; *Montgomery et al., 2016*; *van Dijk et al., 2017*). Insects that are permitted explorative foraging acquire enlarged mushroom bodies compared with constrained siblings, a property also pertaining to the hippocampus (*Basil et al., 1996*; *Montgomery et al., 2016*; *van Dijk et al., 2017*). As is true for mammalian and avian hippocampi voluminous mushroom bodies indicate a species' reliance on spatial or social cues (*Healy and Krebs, 1992*; *Withers et al., 1993*; *Withers et al., 2008*; *Ott and Rogers, 2010*; *Molina and O'Donnell, 2007*; *Molina and O'Donnell, 2008*). Hippocampal lesions impair place memory (*Day et al., 2001*; *Clark et al., 2005*), as do lesions of mushroom bodies (*Mizunami et al., 1998*; *Buehlmann et al., 2020*; *Kamhi et al., 2020*). The volume and neuronal complexity of mushroom bodies in insects also relates to species with highly developed spatial cognition (cockroaches, hymenopterans; *Strausfeld et al., 2009*). The enormous mushroom bodies of varunid crabs suggest comparable cognitive properties relating to space. Shore crabs are opportunistic generalists that live at the interface of two biotopes, marine and terrestrial (*Jacoby, 1981*). They are adept at learning complex tasks (*Tomsic and Romano, 2013*), such as maze learning (*Davies et al., 2019*), organizing 'ad hoc' collaborative actions, establishing social status (*Kaczer et al., 2007*; *Tanner and Jackson, 2012*), and retaining acquired motor skills (*Hughes and O'Brien, 2001*). Operant conditioning and the retention of contextual memories suggest considerable intelligence (*Abramson and Feinman, 1990*; *Pereyra et al., 1999*).

The present description of the varunid mushroom body concludes studies establishing that divergent evolution of the crustacean mushroom body maps to specific malacostracan lineages. These

findings offer hitherto unexplored opportunities for relating divergent cognitive centers to specific ecologies and behavioral repertoires required to negotiate them.

# Materials and methods

## Key resources table

| Reagent type (species) or resource | Designation | Source or reference | Identifiers | Additional information |
|---|---|---|---|---|
| Biological sample | *Hemigrapsus nudus* | Friday Harbor Laboratories | N/A | n = 76 |
| Antibody | α-Tubulin (Mouse, monoclonal) | Developmental Studies Hybridoma Bank (DSHB) | CAT#: 12G10; RRID:AB_1157911 | 1:100 |
| Antibody | α-Tubulin (Rabbit, polyclonal) | Abcam | CAT#: ab15246; RRID:AB_301787 | 1:250 |
| Antibody | Synapsin (SYNORF1; Mouse, monoclonal) | Developmental Studies Hybridoma Bank (DSHB) | CAT#: 3C11; RRID:AB_528479 | 1:100 |
| Antibody | Serotonin (5HT; Rabbit, polyclonal) | ImmunoStar | CAT#: 20080; RRID:AB_572263 | 1:1000 |
| Antibody | Glutamic acid decarboxylase (GAD; Rabbit, polyclonal) | Sigma-Aldrich | CAT#: G5163; RRID:AB_477019 | 1:500 |
| Antibody | Tyrosine hydroxylase (TH; Mouse, monoclonal) | ImmunoStar | CAT#: 22941; RRID: AB_572268 | 1:250 |
| Antibody | Allatostatin (Ast7; Mouse, monoclonal) | Developmental Studies Hybridoma Bank (DSHB) | CAT#: 5F10; RRID:AB_528076 | 1:100 |
| Antibody | DC0 (Rabbit, polyclonal) | Generous gift from Dr. Daniel Kalderon (*Skoulakis et al., 1993*) | RRID:AB_2314293 | 1:400 |
| Antibody | AffiniPure Donkey Anti-Mouse IgG (H+L) Cy3 (polyclonal) | Jackson ImmunoResearch | CAT#: 715-165-150; RRID:AB_2340813 | 1:400 |
| Antibody | AffiniPure Donkey Anti-Rabbit IgG (H+L) Cy5 (polyclonal) | Jackson ImmunoResearch | CAT#: 711-175-152; RRID:AB_2340607 | 1:400 |
| Antibody | AffiniPure Donkey Anti-Rabbit IgG (H+L) Alexa Flour 647 (polyclonal) | Jackson ImmunoResearch | CAT#: 711-605-152; RRID:AB_2492288 | 1:400 |
| Other (serum) | Normal donkey serum | Jackson ImmunoResearch | RRID:AB_2337258 | N/A |
| Other (DNA stain) | SYTO 13 Green Fluorescent Nucleic Acid Stain | Thermo Fisher Scientific | CAT#: S7575 | 1:2000 |
| Other (Phalloidin stain) | Alexa Fluor 488 Phalloidin | Thermo Fisher Scientific | CAT#: 12379; RRID:AB_2315147 | 1:40 |
| Other (Chemical) | α-Terpineol | Sigma-Aldrich | CAT#: 432628 | N/A |
| Other (histology chemical) | Potassium dichromate | Sigma-Aldrich | CAT#: 207802 | N/A |

*Continued on next page*

*Continued*

| Reagent type (species) or resource | Designation | Source or reference | Identifiers | Additional information |
|---|---|---|---|---|
| Other (HPLC purified water) | HPLC Water | Sigma-Aldrich | CAT#: 270733 | N/A |
| Other (histology chemical) | Osmium tetroxide | Electron Microscopy Sciences | CAT#: 19150 | N/A |
| Other (histology chemical) | Ethyl gallate | Fisher Scientific | CAT#: G001625G | N/A |
| Other (histology chemical) | Phosphate buffer tablets | Sigma-Aldrich | CAT#:45ZE83 | N/A |
| Other (histology chemical) | Glutaraldehyde | Electron Microscopy Sciences | CAT#: 16220 | N/A |
| Other (histology chemical) | Paraformaldehyde | Electron Microscopy Sciences | CAT#: 15710 | N/A |
| Other (histology chemical) | Sodium cacodylate | Electron Microscopy Sciences | CAT#: 50-980-232 | N/A |
| Other (histology chemical) | Silver nitrate | Electron Microscopy Sciences | CAT#: 21050 | N/A |
| Other (histology chemical) | Pure copper shot | Sigma-Aldrich | CAT#: 326488 | N/A |
| Other (histology chemical) | Propylene oxide | Electron Microscopy Sciences | CAT#: 20401 | N/A |
| Other (embedding resin) | Durcupan ACM resin (4-part component kit) | Sigma-Aldrich | CAT#: 44610 | N/A |
| Other (embedding medium) | Paraplast Plus | Sigma-Aldrich | CA#: 76258 | N/A |
| Other (mounting medium) | Permount mounting medium | Fisher Scientific | CAT#: SP15-100 | N/A |
| Other (mounting medium) | Entellan | Sigma-Aldrich | CAT#: 1.07961 | N/A |
| Software, algorithm | Photoshop CC | Adobe Inc | N/A | N/A |
| Software, algorithm | Helicon Focus | Helicon Soft | N/A | N/A |
| Software, algorithm | Zen System Software | Zeiss | N/A | N/A |
| Software, algorithm | TrakEM2 | *Cardona et al., 2012* | N/A | N/A |
| Software, algorithm | Amira 2019.4 | Thermo Fisher | N/A | N/A |
| Other (microscope) | LSM 5 Pascal confocal microscope | Zeiss | N/A | N/A |
| Other (microscope) | Orthoplan light microscope | Leitz | N/A | N/A |

## Animals

Crabs (*Hemigrapsus nudus*), with carapace widths between 4 and 6 cm, were collected from designated sites on San Juan Island, Washington, affiliated with the Friday Harbor Marine Biology Laboratory (University of Washington, Seattle). Two further shipments of living crabs were sent to the Strausfeld laboratory and maintained at 15°C in tanks containing seaweed continuously moistened

with salt water. A total of 76 animals were used for this study over an 8 year period. Twenty were used for immunohistology, 8 for osmium-ethyl gallate staining, 10 for Bodian reduced silver staining, and 38 for Golgi impregnations.

## Neuropil staining with osmium-ethyl gallate

A modified ethyl gallate method (*Wigglesworth, 1957*) was used to stain entire brains that were then embedded in plastic and serial sectioned. These were used for Amira 3D reconstructions. After cooling to complete immobility, the eyestalks were opened to expose the lateral protocerebrum. The head was then immediately sliced off behind the brain and with the eyestalks immersed in 0.13 M cacodylate buffered (pH 6.8; Fisher Scientific, Cat#: 50-980-232) 1% paraformaldehyde with 2% glutaraldehyde (Electron Microscopy Sciences, Hatfield, PA). The rest of the animal was placed in a −40°C freezer. After overnight fixation undissected tissue was soaked in buffer. The lateral protocerebra and midbrain were then freed of overlying exoskeleton. Next, tissue was placed for 2 hr in 0.5% osmium tetroxide in cacodylate buffer, continuously rocked and kept at around 4°C in the dark. After a washing in cold buffer, tissue was further well washed in distilled water while raising the temperature to 20°C. Tissue was then immersed in aqueous 0.5% ethyl gallate with constant rocking. Tissue was periodically checked and when dark blue (after about 30 min) thoroughly washed in distilled water, dehydrated, and embedded in Durcupan to be cut serially into 15-μm-thick serial sections using a sliding microtome.

## Neuropil staining with reduced silver

Bodian's original method (*Bodian, 1936*) was used, after fixing tissue in an admixture consisting of 75 parts absolute ethanol, five parts glacial acetic acid, 20 parts 16% paraformaldehyde (Electron Microscopy Sciences; Cat# 16220; Hatfield, PA). As for the osmium-ethyl gallate method, crabs were cooled to complete immobility, the relevant parts removed, opened and fixed in AAF (70 parts absolute ethanol, five parts glacial acetic acid, 25 parts 10% E. M. grade formaldehyde). After 2–4 hr neural tissue was freed under 70% ethanol, then dehydrated to absolute ethanol before clearing in terpineol. It was next soaked in Xylol raising the temperature to 65°C. Finally, tissue was transferred, via a mixture of 50:50 Xylol-Paraplast, into pure Paraplast (Sherwood Medical, St. Louis, MO) held at 65°C in shallow aluminum cookie dishes. After three changes, each about 3–4 min long, the final dish was placed on icy water for the Paraplast to immediately begin to solidify. After serial sectioning at 10 μm, sections were mounted on glycerin-albumin-subbed slides, dried, and dewaxed. Hydrated material was incubated at 60 °C in 2–5% silver proteinate (Roques, Paris) for 24 hr with the addition of 1–4 g pure copper shot/100 ml. Afterward, sections were conventionally treated with hydroquinone, gold toned, fixed, and mounted under Permount (Fisher, Springfield, NJ).

## Silver chromate (Golgi) impregnation

Animals were anesthetized over ice. The midbrain and eyestalk neural tissue was dissected free from the exoskeleton and its enveloping sheath in ice-cold chromating solution comprising 1 part 25% glutaraldehyde (Electron Microscopy Sciences; Cat# 16220; Hatfield, PA), 5 parts 2.5% potassium dichromate (Sigma Aldrich; Cat# 207802; St. Louis, MO) with 3–12% sucrose, all dissolved in high-pressure liquid chromatography (HPLC)-grade distilled water (Sigma Aldrich; Ca#270733). Central brains and their detached lateral protocerebra (tissues) were placed in fresh chromating solution overnight, in the dark at room temperature (RT). Next, tissues were briefly rinsed (30 s with swirling) in 2.5% potassium dichromate and transferred to an admixture of 2.5% potassium dichromate and 0.01% osmium tetroxide (Electron Microscopy Sciences; Cat# 19150; Hatfield, PA) for 24 hr. Tissues were again washed in 2.5% potassium dichromate and immersed for 24 hr in a fresh chromating solution. Before silver impregnation, tissues were decanted into a polystyrene weighing dish, swirled in 2.5% potassium dichromate and then gently pushed with a wooden toothpick into a glass container containing fresh 0.75% silver nitrate (Electron Microscopy Sciences; Cat# 21050; Hatfield, PA) in HPLC water. Tissues were twice transferred to fresh silver nitrate, then left overnight in silver nitrate. Throughout, metal was not allowed to come into contact with tissue. Tissues were rinsed twice in HPLC water. To achieve massive impregnation of neurons, after treatment with silver nitrate tissues were washed in HPLC water and again immersed in the osmium-dichromate solution for 12 hr and then transferred to silver nitrate as described above. After washing in HPLC water and

dehydration through a four step ethanol series from 50% to 100% ethanol tissues were immersed in propylene oxide (Electron Microscopy Sciences; Cat# 20401; Hatfield, PA), and left for 15 min. They were then subject to increasing concentrations of Durcupan embedding medium (Sigma Aldrich; Cat# 44610; St. Louis, MO) in propylene oxide during 3–5 hr. Tissues were left overnight at RT in pure liquid Durcupan and then individually oriented in the caps of BEEM capsules. These were then married to the inverted capsule cut open at its base, and were topped-up with Durcupan. Filled capsules were placed in a 60°C oven for 18–24 hr. for plastic polymerization. Cooled blocks were sectioned at a thickness of 40–50 µm (max) and mounted using Permount mounting medium (Fisher Scientific; Cat# SP15-100; Hampton, NH).

## Immunohistochemistry, antibodies

Visualization of arrangements of immunostained neurons that further define mushroom bodies was achieved using antibodies listed in the key resource table above. An antibody against DC0 that recognizes the catalytic subunit of protein kinase A in *D. melanogaster* (*Skoulakis et al., 1993*) is preferentially expressed in the columnar lobes of mushroom bodies across invertebrate phyla, including members of Lophotrochozoa (*Wolff and Strausfeld, 2015*). Western blot assays of DC0 antibodies used on cockroach, crab, centipede, scorpion, locust, remipede, and millipede neural tissue reveal a band around 40 kDa, indicating cross-phyletic specificity of this antibody (*Stemme et al., 2016*; *Wolff and Strausfeld, 2015*). Antibodies against synapsin (*Klagges et al., 1996*) and α-tubulin (*Thazhath et al., 2002*) were used, often in conjunction with other primary antibodies, to identify dense synaptic regions and general cellular connectivity. Both antibodies likely recognize highly conserved epitope sites across Arthropoda and have been used previously in crustaceans (*Andrew et al., 2012*; *Brauchle et al., 2009*; *Harzsch et al., 1997*; *Harzsch and Hansson, 2008*; *Sullivan et al., 2007*). Antibodies against serotonin (5HT), glutamic acid decarboxylase (GAD), and tyrosine hydroxylase (TH) were used in this study to describe neuronal organization and in distinguishing neuropil boundaries. 5HT is an antibody that has proven to be invaluable for neuroanatomical studies across Arthropoda (*Antonsen and Paul, 2001*; *Harzsch and Hansson, 2008*). Previous studies have used 5HT as a comparative tool for neurophylogenetic analysis (*Harzsch and Waloszek, 2000*). Antibodies against GAD and TH were used to detect the enzymatic precursors of gamma aminobutyric acid (GABA) and dopamine, respectively. These two antibodies do not require the use of alternative fixation methods, making them compatible with synapsin and α-tubulin labeling, while avoiding the need to use glutaraldehyde. Comparisons of anti-GAD and anti-TH immunolabeling with that of their respective derivatives have demonstrated that these antibodies, respectively, label putative GABAergic and dopaminergic neurons (*Cournil et al., 1994*; *Crisp et al., 2002*; *Stemme et al., 2016*; *Stern, 2009*).

## Immunohistochemistry, application

Methods follow those used for two recent studies on the eumalacostracan brain (*Sayre and Strausfeld, 2019*; *Strausfeld et al., 2020*). Animals were anesthetized to immobility using ice. Brains detached from nerve bundles leading to the eyestalks were first dissected free and the eyestalks were then removed. Immediately after removal, tissue was immersed in ice-cold fixative (4% paraformaldehyde in 0.1 M phosphate-buffered saline (PBS) with 3% sucrose [pH 7.4]). Midbrains and lateral protocerebra with their intact optic lobes were desheathed and left to fix overnight at 4°C. Next, tissue was rinsed twice in PBS before being transferred to the embedding medium (5% agarose with 7% gelatin) for 1 hr. at 60°C before cooling to room temperature in plastic molds. After solidification, blocks were removed from the molds and postfixed in 4% paraformaldehyde in PBS for 1 hr. at 4°C. The blocks were then rinsed twice in PBS and sectioned at 60 µm using a vibratome (Leica VT1000 S; Leica Biosystems, Nussloch, Germany). Next, tissue sections were washed twice over a 20 min. period in PBS containing 0.5% Triton-X (PBST). Tissue was subsequently blocked in PBST with 0.5% normal donkey serum (NDS; Jackson ImmunoResearch; RRID:AB_2337258) for 1 hr before primary antibody incubation. Primary antibodies were added to the tissue sections at dilutions listed in Table 1. Sections were left overnight on a rotator at room temperature. Sections were next rinsed in PBST six times over the course of an hour. Donkey anti-mouse Cy3 and donkey anti-rabbit Cy5 or Alexa 647 (Jackson ImmunoResearch; RRID:AB_2340813; RRID:AB_2340607; RRID:AB_2492288, respectively) IgG secondary antibodies were added to Eppendorf tubes at a concentration of 1:400 and

spun for 12 min at 11,000 g. The top 900 µL of the secondary antibody solution was added to the tissue sections, which were then left to incubate overnight at room temperature on a rotator. For F-actin staining, tissue sections were left to incubate in a solution containing phalloidin conjugated to Alexa 488 (Thermo Fisher Scientific; RRID:AB_2315147) at a concentration of 1:40 in PBST for 2–3 days with constant gentle agitation following secondary antibody incubation. To label cell bodies, tissue sections were then rinsed twice in 0.1 M Tris–HCl buffer (pH 7.4) and soaked for 1 hr. in Tris–HCl buffer containing 1:2000 of the nuclear stain, Syto13 (ThermoFisher Scientific; Cat# S7575) on a rotator. Next, sections were rinsed six times in Tris–HCl over the course of 1 hr. before being mounted on slides in a medium containing 25% Mowiol (Sigma Aldrich; Cat# 81381) and 25% glycerol in PBS. Slides were covered using #1.5 coverslips (Fisher Scientific; Cat# 12-544E). To verify secondary antibody specificity, primary antibodies were omitted resulting in complete abolishment of immunolabeling. TH immunolabeling required a modified staining procedure with a shorter fixation time as well as antibody incubation in whole unsectioned tissue. Standard fixation times or sectioning the tissue prior to immunostaining resulted in poor or absent labeling as has been described previously (*Cournil et al., 1994*; *Lange and Chan, 2008*). For TH labeling, neural tissue was dissected and fixed in 4% paraformaldehyde in PBS containing 3% sucrose for 30–45 min. Following fixation, neural tissue was rinsed twice in PBS, and then twice in 0.5% PBST over the course of 40 min. Tissue was then transferred to blocking buffer containing 5% NDS in 1% PBST and left to soak for 3 hr. TH primary antibody was next added to the blocking buffer at a concentration of 1:250. To assist antibody permeation, whole tissues were microwave-treated for 2 cycles of 2 min on low power (~80 W) followed by 2 min no power under a constant vacuum. Tissue was subsequently left to incubate in primary antibody solution for 2–3 days and was microwave-treated each day. After primary antibody incubation, tissue was washed with 0.5% PBST six times over the course of 2 hr and then transferred to a solution containing 1:250 Cy3 secondary antibody. Whole mounts were left in secondary antibody overnight on a gentle shaker before being sectioned and mounted as described above. In dual labeling experiments, sectioned tissue labeled with anti-TH was then stained with an additional primary and secondary antibody also as described above.

## Imaging

Confocal images were collected as Tiff files using a Zeiss Pascal five confocal microscope (Zeiss; Oberkochen, Germany). Image projections were made using Zeiss Zen System software. Light microscopy images were obtained using a Leitz Orthoplan microscope equipped with Plan Apochromat oil-immersion objectives (X40, X60, and X100). Series of step-focused optical sections (0.5–1.0 µm increments) were collapsed onto a single plane using Helicon Focus (Helicon Soft; Kharkov, Ukraine). Images were transferred to Adobe Photoshop (Adobe Systems, Inc; San Jose, CA) and processed using the Photoshop camera raw filter plug-in to adjust sharpness, color saturation and luminance, texture and clarity.

## 3D-reconstruction

Serial sections of osmium-ethyl gallate stained eyestalks were imaged using a brightfield Leitz Orthoplan microscope at a pixel scale of 0.5 µm x 0.5 µm. The sections were cut at a thickness of 15 µm using a rotating microtome. To account for compression and error using dry lenses because of the difference in refractive indices between glass/section and air, image sections were digitally adjusted to a voxel size of 0.5 x 0.5 x 24 µm. Additionally, to obtain a better Z-resolution, two snapshots at two focus planes were taken for each section. The resulting images were manually aligned using non-neuronal fiduciaries in the software TrakEM2 (*Cardona et al., 2012*).

Neuropils, cell bodies, and large nerve tracts were segmented using the software Amira (Amira 2019.4; Thermo Fisher Scientific; Waltham, MA; USA). We achieved this using the Segmentation Editor, a tool which enables tracing of image data by assigning voxels to materials (i.e. user defined objects such as neuropils, cell bodies, etc.) using a paintbrush tool. Selected sections were outlined in the XY plane and interpolated across sections. Then, using the aid of the interpolated outlines, selected sections were traced in the XY, YZ and XZ planes to create a scaffold of the region of interest. The function 'wrap' was then used to interpolate a 3D reconstruction of the object constrained by the scaffold. For visualization purposes, the label field containing 'wrapped' labels was interpolated in Z using the 'Interpolate Labels' module, adding an additional three slices for each already

existing slice while maintaining the overall dimension of the reconstruction. This helped to account for the anisotropic Z-resolution (i.e. 'staircase effect'). The 3D model was then visualized using the 'Surface Generator' module, which created a 3D polygonal surface mesh. Images used for the figures were generated using the snapshot function. The globuli cell population was estimated on the basis of spherical globuli cells, each with the average radius of 3 µm, occupying a volume of $2.5074^3$ µm$^3$ calculated from the serially reconstructed globuli cell mass.

## Acknowledgements

The research described here is supported by the National Science Foundation under Grant No. 1754798 awarded to NJS. Our gratitude is once again directed to Daniel Kalderon, Columbia University, New York, for supplying the DC0 antibodies, as he has during the last decade and longer. We thank the staff of the University of Washington's Friday Harbor Marine Laboratories, San Juan, for their unfailing help in obtaining living specimens. Briana Olea-Rowe and Hannah Joy Miller provided expert technical assistance. Students and trainees have, over a number of years, made significant contribution to histological material contributing to this research. We have profited from discussions and advice from Wulfila Gronenberg (University of Arizona) and Frank Hirth (Kings College, University of London) and are indebted to Camilla Strausfeld for her advice, critically discussing versions of the manuscript, suggesting many improvements and meticulously editing the text. We also thank the three reviewers for providing constructively helpful suggestions.

## Additional information

### Funding

| Funder | Grant reference number | Author |
| --- | --- | --- |
| National Science Foundation | 1754798 | Nicholas Strausfeld |

The funders had no role in study design, data collection and interpretation, or the decision to submit the work for publication.

### Author contributions

Nicholas Strausfeld, Conceptualization, Data curation, Formal analysis, Funding acquisition, Validation, Investigation, Methodology, Writing - original draft, Project administration, Writing - review and editing; Marcel E Sayre, Data curation, Formal analysis, Validation, Investigation, Visualization, Methodology, Writing - review and editing

### Author ORCIDs

Nicholas Strausfeld https://orcid.org/0000-0002-1115-1774
Marcel E Sayre https://orcid.org/0000-0002-2667-4228

### Decision letter and Author response

Decision letter https://doi.org/10.7554/eLife.65167.sa1
Author response https://doi.org/10.7554/eLife.65167.sa2

## Additional files

### Supplementary files

• Transparent reporting form

### Data availability

Source data are preserved on vibratome immunostained sections maintained at 4˚ C in investigator's laboratory (NJS). Likewise, original data of selectively impregnated neurons (Golgi material) and silver stains are curated and maintained in the NJS laboratory.

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
