## [Decision Letter]

**Acceptance summary:**

Arthropod mushroom bodies are high-order integrating centers involved in learning and memory. In this paper Strausfeld and Sayre analyze the detailed architecture of the mushroom body in a shore crab (Brachyura) as the most recent chapter of a comprehensive treatise on crustacean brain evolution. They show how this architecture is very different from that present in many other arthropod groups; providing a clear example of how anatomical structures can vary in different animals to better cope with their specific lifestyles. The emerging picture of homology and divergence of crustacean mushroom bodies is of substantial value for two reasons: 1) it provides a solid framework for continued evolutionary and functional studies at the molecular, developmental, and behavioral levels. 2) comparative neuroscience is critically important for accurate generalization of findings in genetic model species that currently dominate in neuroscience research.

**Decision letter after peer review:**

Thank you for submitting your article "Shore crabs reveal novel evolutionary attributes of the mushroom body" for consideration by *eLife*. Your article has been reviewed by three peer reviewers, and the evaluation has been overseen Ronald Calabrese as the Senior and Reviewing Editor. The following individuals involved in review of your submission have agreed to reveal their identity: Sarah Farris (Reviewer #1); Pedro Martinez Serra (Reviewer #2); Charles Derby (Reviewer #3).

The reviewers have discussed the reviews with one another and the Reviewing Editor has drafted this decision to help you prepare a revised submission.

Summary:

Arthropod mushroom bodies are high-order integrating centers involved in learning and memory. In this paper Strausfeld and Sayre analyze the detailed architecture of the mushroom body in a shore crab (Brachyura) as the most recent chapter of a comprehensive treatise on crustacean brain evolution. They show how this architecture is very different from that present in many other arthropod groups; providing a clear example of how anatomical structures can vary in different animals to better cope with their specific lifestyles. The emerging picture of homology and divergence of crustacean mushroom bodies is of substantial value for two reasons: 1) it provides a solid framework for continued evolutionary and functional studies at the molecular, developmental, and behavioral levels. 2) comparative neuroscience is critically important for accurate generalization of findings in genetic model species that currently dominate in neuroscience research.

Minor points:

The reviewers have identified a number of minor concerns that must be addressed before publication.

Reviewer #1:

I have no major comments or criticisms, only a few suggestions as listed below.

Throughout: I am curious about the rostral, proximal cells that are DC0 positive and the proximal gyrus that is DC0 +AT positive. Could they be associated with each other? Although they are designated MBONs I am skeptical as MBONs are typically not DC0 positive. They are reminiscent of Y-tract Kenyon cells in the Lepidoptera- cell bodies separate from main mass of granule cells, possibly producing a separate neuropil with their dendrites (if proximal gyrus is indeed associated with them).

Figure legends, figures, and text sometimes do not match up, for example:

Results: describe the pedestal of the reniform body but it is not indicated in either figure of Figure 1—figure supplement 1;

The Results describe projection neurons to the LPC in the AGT, which is labeled as the OGT in the figure and legend;

Another pass through the figures/legends/text would clear up any other discrepancies.

Introduction: what is a Hebbian network? Is any network that is capable of associative synaptic plasticity a Hebbian network?

Introduction, eighth paragraph: remove "of".

Add "the" "…ecological topographies at the interface…"

Subsection “The varunid lateral protocerebrum”, second paragraph: replace "is" with "are".

Add "and" "…situated laterally and constrained to a volume".

Subsection “The varunid lateral protocerebrum”, last paragraph: Confusing phrasing suggests that mushroom body axons are carried to the midbrain via the eyestalk nerve (I assume this is referring to efferents).

"Salient both in osmium ethyl gallate stained brains" The other part of both is not indicated (I assume referring to reduced silver staining).

Subsection “Neuronal organization of the calyces”, fourth paragraph: replace "of" with "in".

Space between “calyx 2”.

Unclear phrase "at a level limited to their emergence from the calyces a small part of their initial length".

Merge sentences "difficult to discern in reduced silver, in part because…".

Figure 9 legend: correct spelling (supplying).

Subsection “The reniform body”, last paragraph: is DC0/PKA expression sufficient to define a learning and memory center?

Discussion: Unclear what pyramidal-like means; are they similar to pyramidal cells (?) in morphology, connectivity some other way?

Discussion: correct spelling "accessory".

Figure 3—figure supplement 1 legend: should be afferent terminals or fibers.

Reviewer #2:

The paper is well written and clear. The experimental details and the description of tracts, neuronal processes, and network connections are exhaustive. The evolutionary discussion is particularly interesting, though I would prefer to have a hypothetical scenario on how the specific structure and position of the brachyuran mushroom body has arisen within the crustacea, or eventually, within the Arthropoda (if the authors can provide any).

There are some minor issues that I have to note:

1) There are several typos (or misspelled words) through the text. Please, give another careful read to the text.

2) The name of the species used does not appear until the end of the Introduction. Please provide this name in the Abstract and early in the Introduction; otherwise, the reader needs to wait unduly to discover which species is the target of the paper.

Other comments pertaining to the general presentation of the work:

While, as mentioned, the anatomical description of the mushroom body is exceptional (the Golgi impregnations are spectacular), one is left with the impression that the "proof" that the structures that are constitutive parts of a mushroom body (a center that integrates sensory information) are functioning as such. Needless to say, homology propositions should (if possible) include structural and functional data.

In this context, it would have been nice to have complementary imaging assays on output responses to specific stimulus; a functional approach taken already in crabs by Maza et al. in 2020. Moreover, the lack of tracing experiments, allowing the visualization of projections to the mushroom body substructures does not allow the identification of the afferent tracks providing the sensory information to it.

Other comments that would be important to address:

The lack of resolution (in general) provided by 3D reconstructions make difficult to appreciate the relative disposition of some structures. I would suggest helping the reader by providing a diagram with the positions of key structures such as: Ca1, Ca2, Ca1Gy1, Ca1Gy2, gct1, and gct2. The diagram should include the medial and rostral axis for orientation. Figure 10 gives a partial solution to the problem, though a diagram would be clearer.

We are told that in Figure 10 there is a "synapsin-actin" labelled section. Is that correct? I cannot see any panel with that staining method; while I do see this method used in Figure 3.

The authors have used previously antibodies against DC0, phosphorylated CaMKII (pCAMKII), and Leo in other animals to identify the mushroom body; why not all of them also here?

The Discussion of the results is exhaustive (I am particularly impressed by the discussion of the gyri system). However, I missed a very specific discussion on the terminology and characteristics of the brachyuran mushroom body. I am thinking specifically of a discussion of the data provided by Maza and co-authors. To the point, a similar study in the crab species *Neohelice granulata* (done by Maza and collaborators) has raised some controversy on the specific nature of the mushroom body and its components in the Brachyura. Since the view of Strausfeld and collaborators is at the center of these polemical interpretations, it would be important that the authors delve a little bit more into their different views about crab's mushroom bodies in this paper.

I find Figure 1—figure supplement 1 to be a good illustration of the *Hemigrapsus* brain, and thus, suggest that the authors incorporate it into the article text, perhaps as an introductory diagram to the discussion of the results. The incorporation of a *Drosophila* scheme would be particularly useful for those of us that are better acquainted with the fly mushroom body.

All in all, this is an excellent paper on a particularly interesting topic-the evolutionary history of nervous system architectures. Further work into the cellular components (cell type diversity) and the patterning mechanisms that organize the different subdomains should provide us with the necessary insights into the development of the varunid mushroom body.

Reviewer #3:

Add "the": "at the level of neuronal arrangements".

There is something wrong with the sentence "the following description of uses…"

"Stomatopod" should be lower case (unless you want to use "Stomatopoda").

Subsection “The calyces and their origin from globuli cells”, end of second paragraph: this finding of neurogenesis in the LP is interesting. It is barely mentioned, though. Perhaps expand, and discuss? Also, neurogenesis in a similar region was described in the shore crab *Carcinus maenas* by Schmidt, 1997, and Hansen and Schmidt, 2001.

"a small clusters of cell bodies" should be corrected.

"The origin.… are" should be corrected.

"protocerebra" rather than "protocebra".

References:

In the references, there are many species names that should be in italics;

Spelling should be "Animal Behaviour";

Correct the spelling of "Neuroscience".

Figure 1—figure supplement 1 legend:

"antennular nerve" rather than "antennule nerve" – for parallel construction with "antennal nerve" (rather than (antenna nerve);

Shouldn't "medial" be "medina"?;

Add the missing "(";

"divides" should be "divide".

Figure 3—figure supplement 1 legend: "Afferent terminals fibers" should be corrected.

Figure 4—figure supplement 1 legend: "intrinsic neurons fields" should be corrected.

Figure 7—figure supplement 1 legend: "neuron" should be "neurons".

---

## [Author Response]

The reviewers have identified a number of minor concerns that must be addressed before publication.

We have adopted the changes required, and we have followed all but one of the suggestions. We have introduced the species name *Hemigrapsus nudus* in the Abstract and early in the Discussion. We have followed the advice of the second reviewer and promoted Figure 1—figure supplement 1 into the main body of the paper where it is now Figure 1. The same has been done for Figure 2—figure supplement 1, which is now Figure 4, responding to the comments of the third reviewer. The existing figures have been renumbered accordingly and references to them in the text are adjusted accordingly. These figure numbers are self-evident and do not appear as red mark-ups although the relevant legends do.

Responding to a question from reviewer 1, at this stage of our studies we are unable to identify further traits that might suggest correspondence of the proximal allatostatin containing gyrus to the Y-tract of lepidopteran mushroom bodies. Concerns regarding proximal large perikarya, many of which are DC0 positive, are appropriate and we have responded to these. Our interpretation of them as MBONs is based on Golgi-impregnated morphologies of neurons originating from that cell cluster, the branches of which appear to spread across gyri associated with the calyces and lobes. DC0-immunoreactive axons have been identified extending from those levels into the eyestalk nerve and we have added an inset showing this in what is now Figure 2. The image suggests that some of these neurons may have the status of efferent neurons. Further exploration is required to determine if this is accurate. In the interest of caution we have changed the putative identity of MBONs in this figure to a more neutral status of DC0-positive cell bodies (DC0cbs). All other corrections requested by reviewer 1 have been made.

We agree with the second reviewer that a future aim of studies on brachyuran (and other crustacean) brains should be to employ calcium imagining of sensory induced activity. This is our intention in the longer term. The entire lateral protocerebrum is accessible for optical imaging, as already shown by Maza et al., 2020, the description of which is cited in the present references. Likewise, there is much to be done with regard to tracing all of the afferent supply to the mushroom bodies and efferent from them, not just of this taxon but also of others that are available for experimentation (at present limited due to current conditions), and our focus on evolutionary divergence. A start on that undertaking by us (Strausfeld et al., 2020) demonstrated the ready accessibility of these lateral brain regions by virtue of their disposition in the eyestalk.

The second reviewer suggested the addition of a line diagram that would clarify some of the 3D reconstructions. On first reading this we were puzzled, as our considerable investment in generating a 3D model was specifically geared to demonstrate the elaborate relationships described here. However, on further scrutiny we realized that a critical layer of what was originally Figure 11 had been accidentally inverted such that the gyri ended up unrelated to their respective calyces. This demonstrated how intently the reviewer must have studied this figure, and the perplexity that it generated. This figure has now been corrected to show the accurate relationships of all parts of mushroom bodies and the accompanying reniform body, as well as their correct medial, rostral, and ventral axes (see the revised illustrations, now Figure 13). A supplementary figure for Figure 13 has been added showing a schematic of the relationship between the two calyces and their cognate gyri.

Reviewer 2 proposed that we further reflect on two papers by Maza and colleagues, working on another varunid crab species, that claim a different set of neuropils constituting the crab’s mushroom body (Maza et al., 2016, 2020, both cited). However, consideration of these differences requires considerable discussion and it has the potential of opening a Pandora’s box of dissenting opinions; and, as remarked by the second reviewer, unnecessary polemics. In the present work we simply state that the identification of a center by Maza et al. corresponds to the center identified by Bellonci in 1882 as the reniform body. Bellonci discovered this in the mantis shrimp, where he also discovered the first example of a crustacean mushroom body. He described the two as distinct. The same center in crabs has, however, been given the identity of a mushroom body by Maza et al., despite Thoen et al., 2019, and others pointing out that this interpretation is insecure: our remark in the discussion of the present work is that “similarity, conjunction, and congruence are established criteria for assessing phenotypic homology (Patterson, 1988).” The present work distinguishes the reniform body from the mushroom body. In other lineages, where mushroom bodies accompanied by reniform bodies have been identified Patterson’s criteria exclude the latter as a mushroom body homologue (expressio unius; exclusio alterius). This is unlikely to put the matter to rest, however, and the box will eventually have to be opened as a separate commentary that will include those other lineages.

We know from pleasing experience that the third reviewer is meticulous in inspecting manuscripts and we thank him for his scrutiny of this one; and, also for pointing out two important papers of adult neurogenesis in crabs. Reference to these has been added to the Results section and both references have contributed to remarks in the Discussion.